# Living donors kidney transplantation and oxidative stress: Nitric oxide as a predictive marker of graft function

Djamila Izemrane[1,2☯]*, Ali Benziane[3☯], Mohamed Makrelouf[4‡], Nacim Hamdis[5‡], Samia Hadj Rabia[1,6‡], Sofiane Boudjellaba[2,7‡], Ahsene Baz[1☯], Djamila Benaziza[1☯]

1 Laboratory of Biology and Animal Physiology, Higher Normal School, Kouba, Algiers, Algeria, 2 National Higher Veterinary School, Issad Abbes, Oued Smar, Algiers, Algeria, 3 Department of Nephrology-Hemodialysis and Transplantation, Lamine Debaghine University Hospital, Bab El Oued, Algiers, Algeria, 4 Central Biology Laboratory, Lamine Debaghine University Hospital, Bab El Oued, Algiers, Algeria, 5 Laboratory of Food Technology Research, Faculty of Engineering Sciences-University M'Hamed Bougara, City Frantz Fanon, Boumerdes, Algeria, 6 Department of Nuclear Applications, Nuclear Research Center, Sebala, Algiers, Algeria, 7 Laboratory of Research Management of Local Animal Resources (GRAL), National Higher Veterinary School, Issad Abbes, Oued Smar, Algiers, Algeria

☯ These authors contributed equally to this work.
‡ MM, NH, SHR and SB also contributed equally to this work.
* d.izemrane@ensv.dz

**Data Availability Statement:** We have made our Data on Dryad public. The DOI and URL assigned to us are shown below. DOI: 10.5061/dryad.kkwh70sb3 URL: https://datadryad.org/stash/

## Abstract

### Background

Glomerular filtration rate is the best indicator of renal function and a predictor of graft and patient survival after kidney transplantation.

### Methods

In a single-centre prospective analysis, we assessed the predictive performances of 4 oxidative stress biomarkers in estimating graft function at 6 months and 1 year after kidney transplantation from living donors. Blood samples were achieved on days (D-1, D1, D2, D3, D6 and D8), months (M1, M3 and M6) and after one year (1Y). For donors, a blood sample was collected on D-1. Malondialdehyde (MDA), nitric oxide (NO), glutathione s-transferase (GST), myeloperoxydase (MPO), and creatinine (Cr) were measured by spectrophotometric essays. The estimated glomerular filtration rate by the modification of diet in renal disease equation (MDRD-eGFR) was used to assess renal function in 32 consecutive donor-recipient pairs. Pearson's and Spearman's correlations have been applied to filter out variables and covariables that can be used to build predictive models of graft function at six months and one year. The predictive performances of NO and MPO were tested by multivariable stepwise linear regression to estimate glomerular filtration rate at six months.

### Results

Three models with the highest coefficients of determination stand out, combining the two variables nitric oxide at day 6 and an MDRD-eGFR variable at day 6 or MDRD-eGFR at day 21 or MDRD-eGFR at 3 months, associated for the first two models or not for the third model

share/2amXBXayUq0oPVI2xg_IoF9_
04VM3u7AFdfjfNtymFE.

**Funding:** This article was written as part of a doctoral thesis. The funding comes from state institutions, namely the laboratory of biology and animal physiology of Higher Normal School of Kouba, Algiers, Algeria and the central biology laboratory of Lamine Debaghine University Hospital, Bab El Oued, Algiers, Algeria and personnal funding from Djamila Izemrane and Nacim Hamdis.

**Competing interests:** The authors have declared that no competing interests exist.

with donor age as a covariable (P = 0.000, $r^2$ = 0.599, $r^2$adj = 0.549; P = 0.000, $r^2$ = 0.548, $r^2$adj = 0.497; P = 0.000, $r^2$ = 0.553, $r^2$adj = 0.517 respectively).

## Conclusion

Quantification of nitric oxide at day six could be useful in predicting graft function at six months in association with donor age and the estimated glomerular filtration rate in recipient at day 6, day 21 and 3 months after transplantation.

## Introduction

Kidney transplantation is recognized as the best renal replacement therapy for patients with end-stage renal disease (ESRD). It restores kidney function and improves quality and length of life [1]. The expected function after kidney transplantation depends on many variables: donor features (age, gender, kidney size, and associated comorbidity), harvesting conditions (beating vs. non-heart beating donors, cold ischemia time), and recipient characteristics (warm ischemia time, HLA matching, number of transplants, race, and immunosuppresion) [2]. These parameters are better controlled in living-donor kidney transplantation. Glomerular Filtration Rate (GFR) is considered the best indicator of renal function used to quantify the activity of the kidney and the effectiveness of renal replacement therapy. In addition, clinical events and post-transplant renal function in the first year predict long-term graft survival [3]. Assessment of the GFR is thus necessary for the monitoring of patients after receiving a kidney transplant. Serum creatinine remains the most widely used endogenous marker to estimate GFR in practice [4]. However, the accurate estimation of GFR by creatinine depends on factors such as weight, age, gender and/or race. Several formulas have therefore been constructed to correct the influences of these parameters [5–7]. Some of these equations have been evaluated in renal transplant patients, and the most commonly used are the Modification of Diet in Renal Disease (MDRD), Cockcroft-Gault, and (Chronic Kidney Disease Epidemiology Collaboration) CKD-EPI [7–9].

During the transplantation surgery, the kidney is submitted to blood flow arrest, followed at the time of reperfusion by a sudden increase in oxygen supply. This essential clinical protocol causes massive oxidative stress, which causes tissue damage and cell death [10]. Oxidative stress is a complex phenomenon resulting from the imbalance of cell homeostasis between prooxidants and antioxidants. It is directly caused by reactive species mainly reactive oxygen species (ROS). Some of them contain unpaired electrons called free radicals, whereas others do not, but all of them are very reactive and cause the peroxidation of proteins, carbohydrates, lipids and nucleic acids [11]. Oxidative stress and ROS generation in the kidney disrupt the excretory function of each section of the nephron. It impairs water-electrolyte and acid-base balance and affects kidney regulatory mechanisms [12]. Oxidative stress is directly linked to podocyte damage, depressed glomerular filtration rate, proteinuria, and tubulointerstitial fibrosis [13]. Furthermore, oxidative stress is also related to endothelial cell dysfunction and plays a critical role in chronic kidney disease progression [14]. At this stage, nitric oxide (NO), which is involved in several biological processes, including vasodilatation in smooth muscle cells, inflammation, and immune responses, plays a crucial role [15]. It has been shown that microvascular dysfunction in oxidative stress kidney damage is mediated through nitric oxide synthase (NOS). This event can lead to an impairment of the renal afferent arteriole autoregulation [16], increase in perfusion pressure, causing increases in the amount of superoxide radical ($O^-_2$) [17]. Previous studies have demonstrated that the level of oxidative stress markers

correlates significantly with the level of renal function and increases as chronic kidney disease progresses [18]. Transplanted kidneys are also subject to oxidative stress damage due to pre- and post-transplant conditions that cause reperfusion injury or an imbalance between oxidants and antioxidants [19]. Nevertheless, kidney transplantation seems to restore a nearly normal level of glycoxidative stress markers, but a complete remission is only possible when the renal function is normal [20]. This can be explained by the fact that renal proximal tubules contain many mitochondria which are critical for the energy demanding process of reabsorption of water and solutes. Mitochondria are the largest producers of oxygen radicals, which in turn, increase the susceptibility of kidneys to oxidative stress-induced damage [21]. Thus, the measurement of oxidative stress markers is promising for predicting future risk of graft dysfunction. Despite a significant amount of literature on oxidative stress in relation to renal disease or ischemia/reperfusion in kidney transplant, data regarding the use of oxidative stress biomarkers for prediction of kidney graft function remain limited.

To our knowledge, only two studies have investigated the use of oxidative stress biomarkers for prediction of kidney graft function. In 2014, a study concluded that malondialdehyde level on day 7 might represent a useful predictor of one-year serum creatinine [22]. The second was conducted in 2021 and highlighted the association of higher levels of free thiols at day 1 and day 5 with higher measured GFR at Day 5 as well as with measured GFR at one year [23].

Our study aims to examine, within one year, the variations of systemic levels of four oxidative stress biomarkers, malondialdehyde, nitric oxide, glutathione s-transferase and myeloperoxidase in renal transplant recipients from living donors. It also aims to investigate the association of any of these biomarkers with graft function at six months and one year after transplantation as well as assess their predictive performance in eGFR by MDRD equation at six months and one year post-transplantation.

## Materials and methods

### Study design and patient population

Between october 2017 and november 2019, patients who received consecutive ABO compatible renal transplants at the Nephrology-Hemodialysis and Transplant department of the Lamine DEBAGHINE University Hospital, were recruited, as well as their living related donors and spouses. Exclusion criteria were surgical complications and patients under the age of 18 years. The ethics committee of the Lamine DEBAGHINE University Hospital has approved the study. All patients of the cohort signed a written informed consent. The left kidney was transplanted and the type of anastomosis is uretero-vesical in most cases. Celsior is used as the organ preservation fluid.

### Data collection

At the time of recruitment, demographic, clinical, immunological, serological data, as well as the data related to kidney transplanatation were collected. Data on the evolution of renal function and posttransplant complications were recorded, during the first year.

### Sampling and laboratory

For a total of 10 samples per patient, blood samples were processed as follows: 24 hours before transplant surgery (D-1); on the following morning (12–18) hours after graft reperfusion (D1); then on days 2, 3, 6, 8 and 21(D2, D3, D6, D8, D21) and then after months 1, 3, 6 (M1, M3, M6) and finally 1 year (1Y) after transplantation. For the donors, a sample was collected on D-1. The blood samples were taken by conventional procedures. Blood was centrifuged and the

plasma/serum was aliquoted and frozen at -20°C until further assay. Concentrations in peripheral blood of four oxidative stress biomarkers: plasma malondialdehyde (pMDA), serum nitric oxide (sNO), plasma glutathiones-transferase (pGST), serum myeloperoxydase (sMPO), and serum creatinine (sCr) were mesured by spectrophotometric essays.

### Reagents

For this study, we used high quality analytical chemicals from Sigma (St. Louis, MO).

### Plasma malondialdehyde (pMDA) level measurment

Lipid peroxidation level in plasma was estimated by determining the end product of lipid peroxidation, MDA, by using thiobarbituric acid (TBA) test [24]. An aliquot of 100 μL was added to a reaction mixture containing 50 μL of 8.1% sodium dodecyl sulfate, 375 μL of 20.0% acetic acid (pH 3.5), 375 μL of 0.8% thiobarbituric acid. Samples were then boiled for 1h at 95°C and centrifuged at 3000 g for 10 min. The plasma MDA concentration is calculated using the molar extinction coefficient of the MDA-TBA complex at 532 nm of $1.56 \times 10^5$ mmol. L-1.cm-1. Values are expressed in micromoles per litre (μmol/L).

### Serum nitric oxide (sNO) level determination

The rates of NO were evaluated by the quantification of its stable physiological metabolites (nitrite) [25]. By means of a microplate reader spectrophotometer, the level of nitrite in all serum was determined on the basis of the Griess reaction. A volme of 25 μL of each sample was mixed with 25 μL of Griess reagent (5% sulfanilamide, 0.5% napthylethylenediamine dihydrochloride, and 20% HCl). The samples were incubated at room temperature, protected from light for 20min and the optical density was measured at 543nm. The nitrite concentration was calculated using standard range from a 1 mM sodium nitrite stock solution and expressed in micromoles per litre (μmol/L).

### Serum myeloperoxidase (sMPO) activity

The method based on the O-dianisidine in the presence of $H_2O_2$ is used to assess plasma myeloperoxidase (MPO) activity. A volume of 100 μL of each sample was mixed with 2900 μL of phosphate buffer (50 mM, pH = 6) containing 0.167 mg/mL O-dianisidine Dihydrochloride and 0.1% hydrogen peroxide ($H_2O_2$) [26]. The absorbance of each sample was measured every minute and then for 3 min at λ = 470 nm. One unit of MPO activity corresponds to one micromole of hydrogen peroxide ($H_2O_2$) degraded per min and at 25°C [27]. The following formula is used to estimate the activity of MPO.

MPO (U/L) = (ΔA/min x 3000 μL x $10^6$ μmol/mol) / (11300 L x $mol^{-1}$ x $cm^{-1}$ x 1 cm x 100 μL).

= Δ A/min x 2832 μmol/min

= Δ A/min x 2832 U/L.

Δ A/min = A3 –A2 / 2, A3 = absorbance at three min at 470 nm and A2 = absorbance at two min at 470 nm.

11300 L $mol^{-1}$. $cm^{-1}$ = molar absorptivity coefficient.

Sample volume: 100μL

Total volume: 3000μL

### Plasma glutathione S-transferase (pGST) activity

We mesured the activity of GST using the method of a previous study [28]. Briefly, an aliquot of 100 μL of plasma was diluted into a final volume of 1mL containing 1mM GSH, 1mM

CDNB in 0.1M potassium phosphate buffer, pH 6.5. The optical density was read at 30 second intervals for 3 minutes, starting from the 30[th] second. The enzyme activity was expressed in micromol glutathione oxidation per minute at 25˚C and was calculated using a molar extinction coefficient of 9.6 mM- 1 cm- 1 at 340 nm wavelength.

## Serum creatinine level (sCr) measurment and estimation of graft function

Serum creatinine was measured with the COBAS INTEGRA® 400. To estimate glomerular filtration rate, a simplified modification of diet in renal disease equation (MDRD-eGFR) was used, based on the measure of serum creatinine [7].

For man $MDRD_{ml/min/1,73m}{}^2 = 186 \times (sCr_{mg/dl})^{-1,154} \times (age_{yrs})^{-0,203} \times 1,212$ if African origin
For woman $MDRD_{ml/min/1,73m}{}^2 = 186 \times (sCr_{mg/dl})^{-1,154} \times (age_{yrs})^{-0,203} \times 0.742 \times 1,212$ if African origin

## Statistics

We analysed the normal distribution of continuous variables using the Shapiro-Wilk test. Mean and standard deviation were used to describe the distributions of concentrations and enzymatic activities of serum creatinine (sCr), serum myeloperoxidase (sMPO), serum nitric oxide (sNO), plasma malondialdehyde (pMDA) and plasma glutathione S-transferase (pGST) in the study subjects. The median and the 25th and 75th quartiles (interquartile range [IQR]) were used for variables with skewed distributions. Categorical variables were recoded into binary variables. Depending on the conditions of application, the Student's test or the Wilcoxon-Mann-Whitney tests were performed, to investigate the association between oxidative stress markers and demographic/clinical, treatment variables and to analyse longitudinal changes in oxidative stress marker levels. Pearson's and Spearman's correlations were analysed, respectively, for variables with a normal distribution and for those without, to filter out variables (levels of MDA and NO, activity of MPO and GST, MDRD-eGFR) and covariables (living donor status, recipients and donor's age, recipient's and donors' body mass index (BMI), age and BMI ratio between recipient and donor, recipient and donor gender, pretransplant time on dialysis, warm and cold ischemia, antithymocyte globulin (ATG), primary immunosuppressive treatment, complications, HLA mismatch, delayed graft function (DGF) and acute rejection episodes) that can be used to build a six months and one year predictive model of graft function, as well as to avoid collinearity.

Multivariable stepwise linear regression was performed to assess the predictive performance of NO and MPO in estimating graft function at six months and also to predict graft function at one year.

Statistical analyses were performed using SPSS version 20.0 statistical software. A p-value of $< 0.05$ was considered significant.

## Results

### Study cohort

Between October 2017 and November 2019, 46 adult's recipients were consecutively recruited. Three patients had primary graft failure and had grafts removed, one patient died from a myocardial infarction, and six patients will be excluded due to poor adherence to the sampling schedule. Among the remaining 36 patients, 4 had surgical complications, which limited the study cohort to 32 recipients and their donors. All patients were on standard immunosuppressive therapy consisting of calcineurin inhibitors (cyclosporine or tacrolimus), antiproliferative agents (mycophenolate mofetil) and corticosteroids (prednisone). Calcineurin inhibitors are

prescribed according to the number of mismatches. We reserve tacrolimus for patients with more than 4 mismatches and the presence of DSA. Cyclosporine is prescribed for patients with less than 3 mismatches. The therapeutic ranges are codified by the laboratory in accordance with international standards.

Doses are then adjusted according to the therapeutic range after three months. If non-DSA (donor specific antibody) anti-HLA antibodies are present, calineurin inhibitors are started on D-5. Prednisone is given in full doses of 1 mg/kg, and then tapered from D7 post-transplant. The dose of mycophenolate mofetil is 2 g/d. Table 1 describes the demographic, clinical, immunological, transplantation data and therapeutic characteristics of the recipients and their living donors.

## Oxidative stress biomarkers

**Comparison with healthy subjects.** We compared oxidative markers rates measured in 32 kidney transplanted patients at D-1 with those of their 32 living donors considered as healthy subjects (control group).

The patients and the controls were in the same age category. Before kidney transplatataion, the recipients presented a significantly increased mean±SD pMDA levels (15.76±6.93 vs. 10.28 ±6.78 μmole/L, $P = 0.003$) compared with controls. Despite high values of mean±SD sNO in pre-transplanted recipients compared to control group, the difference was not significant (65.53±57.96 vs. 42.50±24.14 μmole/L, $P = 0.27$). No significant differences were detected in pGST and sMPO activities in pre-transplanted recipients compared to the control group (57.49±18.81 vs. 61.65±20.11 μmole/min, $P = 0.43$ and 59.75±41.01 vs. 42.01±15.01 U/L, $P = 0.29$ respectively).

**Longitudinal change in oxidative stress biomarkers.** Table 2 illustrates the evolution of oxidative stress biomarkers and creatinine during the first year posttransplantation. In six months posttransplantation, the activity of the two enzymes sMPO and pGST did not express any significant fluctuations, except for sMPO activity at D1 compared to pre-transplantation which displayed a significant increase (30.96±26.21 vs. 59.75±41.01 U/L, $P = 0.017$ respectively). Means±SD of pMDA and of sNO significantly decreased on the first day compared to pre-transplantation (15.76±6.97 vs. 10.18±6.02 μmole/L, $P = 0.001$ and 65.53±57.96 vs. 28.34 ±22.54 μmole/L, $P<0.0001$ respectively). A reduction of approximately 35% and 32% in pMDA and sNO rates was respectively observed at six-month post-transplantation.

**Construction of prediction models of graft function.** Pearson's and Spearman's correlations have been applied to filter out variables and covariables that can be used to build predictive models of graft function at six months and one year, as well as to avoid collinearity between them. To assess graft function, MDRD-eGFR based on the measure of serum creatinine was used. Correlations between oxidative stress markers and renal function at six months and one year are represented in Table 3. Out of the four biomarkers, only rates of NO at day 6 and at day 21 as well as MPO activity at one month were correlated to the level of MDRD-eGFR at six months ($P = 0.020$, ρ = 0.416; $P = 0.019$, ρ = 0.441 and $P = 0.017$, ρ = 0.454 respectively). None of the four biomarkers were correlated with MDRD-eGFR at one year. The rates of NO and activity of GST are correlated most of the time with each other, levels of MDA and MPO activity were less correlated. Table 4 shows that most of the MDRD-eGFR variables in recipients are correlated with graft function at six months and one-year post-transplantation. In addition, the MDRD-eGFR variable in donors is correlated with the MDRD-eGFR variable in recipients at six months. Correlations between demographic/clinical characteristics, treatment and data related to kidney transplantation with renal function at six months and one year are illustrated in Table 5.

**Table 1. Demographic, clinical, immunological and therapeutic characteristics in kidney transplant donors and recipients.**

| | |
|---|---|
| **Donors** | |
| Age (yr) | 39.8±10.6 |
| Male sex | 11/32 (34.4%) |
| BMI (kg/m$^2$) | 27.2±3.5 |
| Serum creatinine (mg/l) | 7.0±1.6 |
| MDRD-eGFR (ml/min/1,73m$^2$) (non Africain) | 117.8±28.8 |
| **Recepients** | |
| Age (yr) | 35.5±11 |
| Male sex | 26/32 (81.25%) |
| BMI (kg/m$^2$) | 23.7±4.5 |
| Serum creatinine (mg/l) | 91.8±23.4 |
| MDRD-eGFR (ml/min/1,73m$^2$) (non Africain) | 7.2±1.9 |
| Time on dialysis (mo) | 36.7±45 |
| Cause of kidney disease | |
| Indeterminate | 25/32 (78.125%) |
| IgA nephropathy | 2/32 (6.25%) |
| Chronic glomerulonephritis | 2/32 (6.25%) |
| Tubulointerstitial nephritis | 1/32 (3.125%) |
| hypertensive nephropathy | 1/32 (3.125%) |
| Lupus nephropathy | 1/32 (3.125%) |
| Induction regimen | |
| Antithymocyte globulin (ATG) | 28/32 (87.5%) |
| Basiliximab | 4/32 (12.5%) |
| Immunosuppression at time of discharge | |
| Cyclosporine A+ Mycophenolate mofetil+ prednisone | 19/32 (59.37%) |
| Tacrolimus + Mycophenolate mofetil+ prednisone | 13/32 (40.63%) |
| Complications | 18/32 |
| Rejection | 6/18 |
| CMV infection | 6/18 |
| Recurrence of membrano-proliferative glomerulonephritis | 1/18 |
| Giardiosis infection | 1/18 |
| DGF | 4/18 |
| **Donors-Recepients** | |
| HLA mismatch (A, B, DR) | 3 (2–3) |
| Warm ischemia (sd) | 128±88.5 |
| Cold ischemia (min) | 102.2 ±27.6 |
| Age Donor/Age Recipient | 0.95±0.3 |
| BMI Donor/BMI Recipient | 0.89±0.2 |
| Related living donor | 25/32 (78.1%) |
| Brother or Sister | 14/32 (43.75%) |
| Parents | 7/32 (21.9%) |
| Children | 1/32 (3.1%) |
| Aunts and uncles | 2/32 (6.25%) |
| Cousins | 1/32 (3.1%) |
| Non related living donor (partner) | 7/32 (21.9%) |

Note: Values are expressed as mean ±standard deviation, absolute numbers and percentages or median (interquartile range).

HLA, human leucocyte antigen; BMI, body Mass Index; MDRD-eGFR, estimated glomerular filtration rate by modification of diet in ranal disease equation; CMV, cytomegalovirus; DGF, delayed graft function.

**Table 2. Temporary variations of oxidative stress biomarkers and creatinine in plasma and serum of recipients and donors during the first year posttransplantation.**

| | pMDA (μmole/L) | sMPO (U/L) | pGST (μmole/min) | sNO (μmole/L) | sCr (mg/L) |
|---|---|---|---|---|---|
| Healthy subjects | 10.28±6.78 | 42.01±15.01 | 61.65±20.11 | 42.50±24.14 | 07.04±01.58 |
| D-1 | 15.76±6.97 | 59.75±41.01 | 57.50±18.81 | 65.53±57.96 | 91.84±23.87 |
| D1 | 10.18±6.02 | 30.96±26.21 | 59.76±20.67 | 28.34±22.54 | 37.75±19.66 |
| D2 | 10.92±7.11 | 40.78±31.83 | 60.77±22.63 | 40.06±34.54 | 24.31±22.17 |
| D3 | 11.55±5.64 | 46.57±35.35 | 63.23±22.10 | 43.30±33.57 | 19.87±20.99 |
| D6 | 11.20±7.20 | 57.66±44.55 | 61.28±26.28 | 52.07±38.31 | 16.53±19.06 |
| D8 | 10.89±6.98 | 57.71±43.59 | 66.02±28.45 | 52.75±43.84 | 15.12±14.04 |
| D21 | 12.19±5.07 | 66.72±44.06 | 65.65±18.84 | 62.46±43.64 | 14.11±05.90 |
| M1 | 13.90±7.19 | 62.57±40.74 | 66.21±21.47 | 54.53±30.78 | 13.58±06.70 |
| M3 | 09.96±6.69 | 50.37±36.70 | 66.55±22.51 | 45.45±37.05 | 13.10±03.75 |
| M6 | 10.22±5.96 | 49.91±27.56 | 67.84±17.33 | 44.50±35.83 | 13.12±05.19 |
| Y1 | ID | ID | ID | ID | 16.14±18.43 |

Note: Values are expressed as mean ±standard deviation. D, day; M, months; Y, year; pMDA, plasma malondialdehyde; sMPO, serum myeloperoxidase; pGST, plasma glutathione s-transferase; sNO, serum nitric oxide; sCr, serum creatinine; ID: insufficient data.

Donor age was correlated with MDRD-eGFR at six months ($P$ = 0.049, r = -0.351), while donor BMI, complication and warm ischemia were correlated with MDRD-eGFR at one year ($P$ = 0.013, ρ = -0.436; $P$ = 0.010, ρ = -0.450 and $P$ = 0.017, ρ = 0.431 respectively). The results ($P$ and r) of the bivariate correlations shown in Tables 3–5 represent also the results of univariate linear regressions. No collinearity was observed between the variables that were used to build the predictive models.

## Prognosis of graft function at six months

We compared the predictive performances of NO levels at day 6, at day 21 and MPO activity at one month in assessing MDRD-pGFR (predicted glomerular filtration rate by the Modification of diet in renal disease) equation at six months, to the predictive performances of all MDRD-eGFR variables at all times potentially predictive, by applying multivariable stepwise linear regression. Tables 6, 7 shows results of multivariable stepwise linear regression for prognosis of graft function at six months. For each model, the variables introduced and excluded are listed above. In the first step, the variables NO at day 6, NO at day 21, MPO at one month and all of MDRD-eGFR variables were tested separately associated to donor age as covariable (Table 6 models 1–12). In the second step, the combination of one of these variables with one variable of MDRD-eGFR, in addition to donor age were tested. The best combinations with the highest values of regression coefficients "r", coefficients of determination "$r^2$", adjusted coefficients of determination "adjusted $r^2$" are indicated in Tables 6, 7 (Table 7 models 13–15). Both models 2 and 3 show that the NO variables at day 21 and MPO at one month lose their statistical significance and were thus removed from the final model. There is therefore no interest in assessing these variables in the second step. The mathematical equations of the predictive models are indicated in the Table 8 and are of the following form:

$\hat{Y} = \beta_0 + \beta_1 X_1 + \beta_2 X_2 + \beta_3 X_3 + \ldots$

$\hat{Y}$: Predicted variable.

$\beta_0$: Constant.

$\beta_1, \beta_2, \beta_3$: Regression coeifficient of each predictor.

$X_1, X_2, X_3$: Significant predictor variable.

**Table 3. Correlations between oxidative stress markers and renal function at six months posttransplantation.**

| | | GST_D-1 | GST_D1 | GST_D2 | GST_D3 | GST_D6 | GST_D8 | GST_D21 | GST_1M | GST_3M | GST_6M | GST_Donors |
|---|---|---|---|---|---|---|---|---|---|---|---|---|
| MDRD-eGFR_6M | r/ ρ | -0.175 | -0.174 | -0.198 | **0.109** | -0.045 | -0.001 | -0.126 | **-0.109** | **-0.050** | 0.255 | 0.158 |
| | P | 0.374 | 0.368 | 0.285 | **0.558** | 0.807 | 0.995 | 0.522 | **0.573** | **0.788** | 0.190 | 0.431 |
| MDRD-eGFR_1Y | r/ ρ | -0.193 | -0.187 | -0.048 | **0.293** | 0.094 | 0.102 | 0.149 | **0.119** | **0.001** | 0.124 | -0.234 |
| | P | 0.326 | 0.331 | 0.799 | **0.104** | 0.608 | 0.584 | 0.450 | **0.540** | **0.994** | 0.530 | 0.506 |
| | | MPO_D-1 | MPO_D1 | MPO_D2 | MPO_D3 | MPO_D6 | MPO_D8 | MPO_D21 | MPO_1M | MPO_3M | MPO_6M | MPO Donors |
| MDRD-eGFR_6M | r/ ρ | **0.149** | **0.329** | **0.255** | **0.010** | **0.262** | **0.045** | **0.137** | **0.454**[*] | **0.077** | **0.005** | -0.133 |
| | P | **0.467** | **0.093** | **0.166** | **0.956** | **0.148** | **0.810** | **0.487** | **0.017** | **0.687** | **0.979** | 0.509 |
| MDRD-eGFR_1Y | r/ ρ | **-0.160** | **0.178** | **0.048** | **-0.008** | **0.066** | **-0.227** | **-0.052** | **0.025** | **-0.081** | **0.017** | -0.011 |
| | P | **0.435** | **0.375** | **0.799** | **0.965** | **0.718** | **0.218** | **0.793** | **0.900** | **0.672** | **0.928** | 0.241 |
| | | MDA_D-1 | MDA_D1 | MDA_D2 | MDA_D3 | MDA_D6 | MDA_D8 | MDA_D21 | MDA_1M | MDA_3M | MDA_6M | MDA_Donors |
| MDRD-eGFR_6M | r/ ρ | -0.306 | -0.296 | 0.109 | 0.272 | **0.083** | **0.118** | 0.206 | 0.287 | **0.179** | 0.035 | 0.178 |
| | P | 0.113 | 0.119 | 0.567 | 0.153 | **0.664** | **0.534** | 0.302 | 0.138 | **0.345** | 0.860 | 0.374 |
| MDRD-eGFR_1Y | r/ ρ | -0.036 | -0.249 | 0.083 | 0.282 | **0.067** | **-0.135** | -0.002 | 0.238 | **0.099** | -0.243 | 0.019 |
| | P | 0.854 | 0.193 | 0.664 | 0.138 | **0.725** | **0.478** | 0.992 | 0.222 | **0.604** | 0.212 | 0.926 |
| | | NO_D-1 | NO_D1 | NO_D2 | NO_D3 | NO_D6 | NO_D8 | NO_D21 | NO_1M | NO_3M | NO_6M | NO_Donors |
| MDRD-eGFR_6M | r/ ρ | **0.216** | **0.382** | **0.235** | **0.274** | **0.416**[*] | **0.276** | **0.441**[*] | 0.444 | **0.357** | **0.275** | **0.214** |
| | P | **0.280** | **0.054** | **0.212** | **0.143** | **0.020** | **0.133** | **0.019** | 0.058 | **0.053** | **0.149** | **0.275** |
| MDRD-eGFR_1Y | r/ ρ | **-0.049** | **0.093** | **0.087** | **0.093** | **0.324** | **0.016** | **0.179** | 0.134 | **0.101** | **0.032** | **0.006** |
| | P | **0.806** | **0.650** | **0.647** | **0.624** | **0.076** | **0.932** | **0.362** | 0.495 | **0.594** | **0.870** | **0.974** |

Note: values shown in bold type indicate a Spearman correlation and values shown in thin type indicate a Pearson correlation. Results of this table represent also those of simple linear regression.

D, day; M, months; GST, glutathione s-transferase; MDA, malondialdehyde; NO, nitric oxide; MPO, myeloperoxidase; MDRD-eGFR, estimated glomerular filtration rate by the modification of diet in renal disease equation.

P $\leq$ 0.05; ** P $\leq$ 0.01; P*** $\leq$ 0.001.

## Prognosis of graft function at one year

To predict renal function at one year, we included in this model all covariables correlated with one-year MDRD-eGFR rate (donor BMI, warm ischemia and complication), with the MDRD-

**Table 4. Correlations of estimated Glomerular Filtration Rate by Modification of Diet in Renal Disease equation variables with graft function at six months and one year.**

| | | MDRD-eGFR_D-1 | MDRD-eGFR_D1 | MDRD-eGFR_D2 | MDRD-eGFR_D3 | MDRD-eGFR_D6 | MDRD-eGFR_D8 | MDRD-eGFR_D21 | MDRD-eGFR_1M | MDRD-eGFR_3M | MDRD-eGFR_6M | MDRD-eGFR_1Y | MDRD-eGFR Donors_ |
|---|---|---|---|---|---|---|---|---|---|---|---|---|---|
| MDRD-eGFR_6M | r/ ρ | 0.149 | **0.318** | 0.528** | 0.451** | 0.478** | **0.531**[**] | 0.563** | 0.459** | **0.649**[**] | 1 | 0.648** | 0.486** |
| | P | 0.415 | **0.076** | 0.002 | 0.010 | 0.006 | **0.002** | 0.001 | 0.008 | **0.000** | | 0.000 | 0.006 |
| MDRD-eGFR_1Y | r/ ρ | 0.161 | **0.471**[**] | 0.440* | 0.482** | 0.416* | **0.469**[**] | 0.415* | 0.283 | **0.567**[**] | 0.648** | 1 | 0.109 |
| | P | 0.379 | **0.006** | 0.012 | 0.005 | 0.018 | **0.007** | 0.018 | 0.117 | **0.001** | 0.000 | | 0.560 |

Note: values shown in bold type indicate a Spearman correlation and values shown in thin type indicate a Pearson correlation. Results of this table represent also those of simple linear regression.

D, day; M, months; Y, year; MDRD-eGFR, estimated glomerular filtration rate by the modification of diet in renal disease equation.

*P $\leq$ 0.05

** P $\leq$ 0.01

P*** $\leq$ 0.001.

**Table 5. Correlations between demographic/clinical characteristics, treatment and data related to kidney transplantation with renal function at six months and one year.**

|  | MDRD-eGFR_6M | | MDRD-eGFR_1Y | |
|---|---|---|---|---|
|  | $r/\rho$ | P | $r/\rho$ | P |
| Donor age | -0.351* | 0.049 | -0.229 | 0.207 |
| Recipient age | **-0.292** | **0.105** | **0.047** | **0.800** |
| Recipient gender | **0.165** | **0.367** | **-0.208** | **0.253** |
| Donor gender | **0.146** | **0.425** | **0.267** | **0.139** |
| Donor type | **-0.020** | **0.911** | **-0.098** | **0.593** |
| BMI recipient | -0.239 | 0.188 | -0.178 | 0.330 |
| BMI donor | **-0.278** | **0.123** | **-0.436**\* | **0.013** |
| Time of dialysis | **-0.178** | **0.328** | **0.117** | **0.525** |
| Cold ischemia | **-0.307** | **0.106** | **-0.273** | **0.152** |
| Warm ischemia | **-0.030** | **0.875** | **0.431**\* | **0.017** |
| ATG | **-0.251** | **0.166** | **0.056** | **0.760** |
| Primary immunosuppressive treatment | **0.010** | **0.955** | **0.300** | **0.095** |
| Complications | **-0.256** | **0.158** | **-0.450**\*\* | **0.010** |
| DGF | **-0.194** | **0.286** | **-0.138** | **0.451** |
| Rejection | **-0.165** | **0.367** | **-0.304** | **0.091** |
| Recipient age/Donor age | 0.075 | 0.682 | 0.210 | 0.248 |
| Recipient BMI/Donor BMI | 0.102 | 0.579 | 0.076 | 0.679 |
| HLA mismatch (A, B, DR) | **-0.111** | **0.559** | **-0.016** | **0.935** |

Note: values shown in bold type indicate a Spearman correlation and values shown in thin type indicate a Pearson correlation. Results of this table represent also those of simple linear regression. D, day; M, months; Y, year; MDRD-eGFR, estimated glomerular filtration rate by the modification of diet in renal disease equation; ATG, antithymocyte globulin; DGF, delayed graft function; BMI, body mass index.

*P $\leq$ 0.05

** P $\leq$ 0.01

P*** $\leq$ 0.001.

eGFR at six months variable, that expresses the highest value of the correlation coefficients ($r = 0.648$ and $P < 0.01$) among the MDRD-eGFR variables potentially predictive (Table 4). Results of multivariable stepwise linear regression for prognosis of graft function at one-year post-transplantation are indicated in Table 9. The introduced and excluded variables and covariables are listed at the bottom of the table. No variable is transformed.

## Discussion

In our prospective longitudinal study, we set the hypothesis that oxidative stress biomarkers may be predictive of graft function at six months and at one year in living donor transplant recipients. For this purpose, we first measured in recipients the rates of sCr, pMDA, sNO and activities of pGST and sMPO on days (D-1, D1, D2, D3, D6 and D8), months (M1, M3 and M6) and after one year (1Y) post transplantation. We also measured these biomarkers in donors. MDRD-eGFR, based on serum creatinine, was used to assess the renal function of recipients and their donors. Most studies agree that oxidative stress increases progressively with the advanced stages of chronic renal failure. The systemic concentration of oxidising molecules is high among patients with chronic renal failure, and enzymatic activities and the level of antioxidant molecules in the blood are low, leading to oxidative stress [29–31].

The present results show signifiantly elevated levels of MDA, in pre-transplant patients compared to donors, considered to be healthy subjects. This result was in accordance with

**Table 6. Results of multivariable stepwise linear regression for prognosis of graft function at six months.**

| Model | Significant predictor | Regression coeifficient | P | 95% CI | r | $r^2$ | $r^2$ adjusted | P |
|---|---|---|---|---|---|---|---|---|
| 1 | NO D6 | 0.248 | 0.007 | 0.074 - 0.422 | 0.621 | 0.386 | 0.336 | 0.002 |
| | Donor age | - 0.826 | 0.012 | -1.456 - -0.196 | | | | |
| 2 | Donor age | - 0.980 | 0.024 | -1.819- -0.141 | 0.426 | 0.181 | 0.150 | 0.024 |
| 3 | Donor age | - 0.934 | 0.029 | -1.764- -0.103 | 0.420 | 0.177 | 0.144 | 0.029 |
| 4 | MDRD-eGFR D1 | 0.813 | 0.004 | 0.278–1.347 | 0.493 | 0.243 | 0.218 | 0.004 |
| 5 | MDRD-eGFR D2 | 0.494 | 0.002 | 0.198–0.791 | 0.528 | 0.279 | 0.254 | 0.002 |
| 6 | MDRD-eGFR D3 | 0.366 | 0.004 | 0.126–0.605 | 0.586 | 0.343 | 0.298 | 0.002 |
| | Donor age | - 0.882 | 0.019 | -1.607- -0.158 | | | | |
| 7 | MDRD-eGFR D6 | 0.329 | 0.003 | 0.122–0.535 | 0.598 | 0.358 | 0.314 | 0.002 |
| | Donor age | - 0.846 | 0.022 | -1.562- -0.131 | | | | |
| 8 | MDRD-eGFR D8 | 0.392 | 0.001 | 0.175–0.609 | 0.559 | 0.313 | 0.290 | 0.001 |
| 9 | MDRD-eGFR D21 | 0.671 | 0.001 | 0.304–1.037 | 0.639 | 0.409 | 0.368 | 0.000 |
| | Donor age | - 0.716 | 0.042 | -1.405- -0.026 | | | | |
| 10 | MDRD-eGFR M1 | 0.575 | 0.008 | 0.160–0.989 | 0.459 | 0.211 | 0.185 | 0.008 |
| 11 | MDRD-eGFR M3 | 0.662 | 0.000 | 0.413–0.911 | 0.704 | 0.496 | 0.479 | 0.000 |
| 12 | MDRD-eGFR donor | 0.426 | 0.006 | 0.135–0.718 | 0.486 | 0.236 | 0.210 | 0.006 |

Variables introduced in the model:

1 NO D6, donor age and MDRD-eGFR M6 as dependent variable.

2 NO D21, donor age and MDRD-eGFR M6 as dependent variable. NO D21 loses its significant and was removed from the final model statistical

3 MPO M1, donor age and MDRD-eGFR M6 as dependent variable. MPO M1 loses its statistically significant and was removed from the final model.

4 MDRD-eGFR D1, donor age and MDRD-eGFR M6 as dependent variable. Donor age loses its statistically significant and was removed from the final model.

5 MDRD-eGFR D2, donor age and MDRD-eGFR M6 as dependent variable. Donor age loses its statistically significant and was removed from the final model.

6 MDRD-eGFR D3, donor age and MDRD-eGFR M6 as dependent variable.

7 MDRD-eGFR D6, donor age and MDRD-eGFR M6 as dependent variable.

8 MDRD-eGFR D8, donor age and MDRD-eGFR M6 as dependent variable. Donor age loses its statistically significant and was removed from the final model.

9 MDRD-eGFR D21, donor age and MDRD-eGFR M6 as dependent variable.

10 MDRD-eGFR M1, donor age and MDRD-eGFR M6 as dependent variable. Donor age loses its statistically significant and was removed from the final model.

11 MDRD-eGFR M3, donor age and MDRD-eGFR M6 as dependent variable. Donor age loses its statistically significant and was removed from the final model.

12 MDRD-eGFR donor, donor age and MDRD-eGFR M6 as dependent variable. Donor age loses its statistically significant and was removed from the final model.

**Table 7. Results of multivariable stepwise linear regression for prognosis of graft function at six months post-transplantation.**

| Model | Significant Predictor | Regression coeifficient | P | 95% CI | r | $r^2$ | $r^2$ adjusted | P |
|---|---|---|---|---|---|---|---|---|
| 13 | NO D6 | 0.226 | 0.002 | 0.092–0.360 | 0.774 | 0.599 | 0.549 | 0.000 |
| | MDRD–eGFR D6 | 0.339 | 0.000 | 0.169–0.509 | | | | |
| | Donor age | -0.691 | 0.033 | -1.322- -0.061 | | | | |
| 14 | NO D6 | 0.227 | 0.011 | 0.056–0.398 | 0.740 | 0.548 | 0.497 | 0.000 |
| | MDRD–eGFR D21 | 0.591 | 0.001 | 0.259–0.923 | | | | |
| | Donor age | -0.862 | 0.009 | -1.492- -0.231 | | | | |
| 15 | NO D6 | 0.190 | 0.000 | 0.438–1.101 | 0.743 | 0.553 | 0.517 | 0.000 |
| | MDRD–eGFR M3 | 0.770 | 0.010 | 0.050–0.331 | | | | |

Variables introduced in the model:

1 NO D6, MDRD-eGFR D6, donor age and MDRD-eGFR M6 as dependent variable.

2 NO D6, MDRD-eGFR D21, donor age and MDRD-eGFR M6 as dependent variable.

3 NO D6, MDRD-eGFR M3, donor age and MDRD-eGFR M6 as dependent variable. Donor age loses its statistical significant and was removed from the final model.

NO, levels of serum oxide nitric; D, day; M, month; MDRD-eGFR, estimated glomerular filtration rate by modification of diet in renal disease equation.

**Table 8. Mathematical equations of the predictive models (Predicted variable; Ŷ = MDRD-pGFR M6 (ml/min/1.73m$^2$).**

| Models | β0 | β1 | x1 | β2 | x2 | β3 | x3 |
|---|---|---|---|---|---|---|---|
| Model 1 | 90.378 | + 0.248 | NO D6 | - 0.826 | Donor age | / | / |
| Model 2 | 110.934 | - 0.980 | Donor age | / | / | / | / |
| Model 3 | 105.881 | - 0.924 | Donor age | / | / | / | / |
| Model 4 | 51.244 | + 0.813 | MDRD-eGFR D1 | / | / | / | / |
| Model 5 | 46.764 | + 0.494 | MDRD-eGFR D2 | / | / | / | / |
| Model 6 | 82.413 | + 0.366 | MDRD-eGFR D3 | - 0.882 | Donor age | / | / |
| Model 7 | 78.219 | + 0.329 | MDRD-eGFR D6 | - 0.846 | Donor age | / | / |
| Model 8 | 40.492 | + 0.392 | MDRD-eGFR D8 | / | / | / | / |
| Model 9 | 55.542 | + 0.671 | MDRD-eGFR D21 | - 0.716 | Donor age | / | / |
| Model 10 | 32.824 | + 0.575 | MDRD-eGFR M1 | / | / | / | / |
| Model 11 | 24.823 | + 0.662 | MDRD-eGFR M3 | / | / | / | / |
| Model 12 | 21.271 | + 0.426 | MDRD-eGFR donor | / | / | / | / |
| Model 13 | 59.414 | + 0.226 | NO D6 | + 0.339 | MDRD-eGFR D6 | - 0.691 | Donor age |
| Model 14 | 53.818 | + 0.227 | NO D6 | + 0.591 | MDRD-eGFR D21 | - 0.862 | Donor age |
| Model 15 | 8.032 | + 0.190 | NO D6 | + 0.770 | MDRD-eGFR M3 | / | / |

NO, oxide nitric (μmole/L); D, day; M, month; MDRD-eGFR, estimated glomerular filtration rate by modification of diet in renal disease (ml/min/1.73m$^2$).

previous studies [22, 32]. No significant difference between recipients at D-1 (ESRD patients) and controls was noted in NO level, despite a 50% increase over controls in ESRD patients. This can be explained by small sample size, small effect size and large variation in the sample [33]. Our results are similar to previous studies [34, 35]. No significant differences were detected in GST and MPO activities in pre-transplanted recipients compared to the control group. In addition, both enzymes show no significant change during the first year of post-transplant follow-up.

Our results are consistent with a study where there were no significant differences in GST-α levels between donors, which were considered as controls, and recipients before living-donor liver tranplantation [36]. Similarly, plasma MPO protein concentration, measured by ELISA, did not significantly differ between predialysis and control subjects. In contrast, MPO concentration was markedly increased in hemodialysis patients compared to control subjects or predialysis patients [37].

We observed a reduction of approximately 35% and 32% in pMDA and sNO levels respectively six months after transplantation. Our data suggest that the improvement in MDA and NO biomarkers begins on the first day of kidney transplantation and continues throughout

**Table 9. Result of multivariable stepwise linear regression for prognosis of graft function at one year post-trasplantation.**

| Model | Significant predictor | Regression coeifficient | P | 95% CI | r | r$^2$ | r$^2$ adjusted | P |
|---|---|---|---|---|---|---|---|---|
| | Complication | -12.471 | 0.027 | - 23.435- -1.507 | 0.831 | 0.690 | 0.655 | 0.000 |
| | MDRD-eGFR M6 | 0.676 | 0.000 | 0.419–0.934 | | | | |
| | Warm ischemia | 0.101 | 0.002 | 0.040–0.161 | | | | |

Variables introduced in the model: MDRD-eGFR M6, donor BMI, complications, warm ischemia and MDRD-eGFR at Y1 as dependent variable. Donor BMI loses its statistical significance and was removed from the final model prediction.

MDRD-pGFR 1Y$_{(ml/min/1.73m^2)}$ = 18.386 + 0.676 x MDRD-eGFR M6$_{(ml/min/1.73m^2)}$ − 12.471 x complication + + 0.101 x warm ischemia$_{(sd)}$.

A complication = 1; no complication = 0; MDRD-eGFR M6, estimated glomerular filtration rate by modification of diet in renal disease equation at six months; MDRD-pGFR, predicted glomerular filtration rate by the modification of diet in renal disease equation.

the study period. These results are consistent with those of previous studies carried out on cohorts of patients transplanted from living donors, in which plasma levels of MDA and nitrate were analyzed before and for 28 days after transplantation for MDA and before and for 14 days after transplantation for nitrate, respectively [32, 38, 39]. Renal transplantation from living donors rapidly normalised creatinine, urea, GFR, citruline and nitrate. However, despite increased net protein catabolism in peripheral tissues, indicated by increased phenylalanine/ tyrosine molar ratios, low arginine and high asymmetric dimethylarginine concentrations persisted throughout the period examined. Alterations in other amino acids also suggest a similar disturbance in arginine metabolism in recipients after renal transplantation [38]. Furthermore, rates of proinflamatory proteins interleukin 6 (IL-6), tumor necrosis factor alpha (TNF- α) and C-reactive protein (CRP)), as well as those of plasma protein carbonyls and F2-isoprostanes known to be markers of oxidative stress have been reported as being significantly elevated in ESRD, with significant decreases two months after renal transplantation from living donors [40]. On the other hand, a previous study reports that there was no significant change in antioxidant enzyme activities, glutathione peroxidase, catalase and superoxide dismutase during the monitored period of three months in deceased donors [39].

Our results indicate that there are no significant differences in the levels of the four biomarkers between male and female patients, who have had complications and those who have not.

Secondly, we applied Spearman and Pearson bivariate correlations to filter out variables (pMDA level, sNO level, pGST activity, sMPO activity, MDRD-eGFR) and covariables (demographic/clinical characteristics, treatment and renal transplant data related to kidney transplantation) that can be used to build predictive models of graft function (MDRD-pGFR) at 6M and 1Y, as well as to avoid collinearities between all variables. We compared the predictive performances of NO levels at D6, at D21 and MPO activity at 1M in assessing MDRD-pGFR equation at 6M, to the predictive performances of all MDRD-eGFR variables at all times potentially predictive. Both models 2 and 3 show that the NO variables at D 21 and MPO at 1M lose their statistical significances and were removed from the final model. Models of MDRD-pGFR are based on multivariable stepwise linear regression. None of the oxidative stress biomarkers predicted graft function at 1Y.

Our bivariate correlation results, which as a reminder also represent the results of univariate linear regressions at 6M, shows that NO at D6 has a predictive performance close to that of the MDRD-eGFR variable at D3 post-transplantation ($\rho$ = 0.416 and r = 451 respectively) and an increase of the predictive performances of the MDRD-eGFR variables as the sixth month approaches ($r_{D2}$ = 0.528, $r_{D3}$ = 0.451, $r_{D6}$ = 0.478, $\rho_{D8}$ = 0.531, $r_{M1}$ = 0.563, $\rho_{M3}$ = 0,649 respectively). On the other hand, we found that NO levels on day six are also predictive of creatinine levels at six months.

We also found that living donor parameters independently predicted MDRD-pGFR, with donor age predicting graft function at six months (P = 0.049, r = -0.351), while donor BMI predicted graft function at one year (P = 0.013, $\rho$ = -0.436). Among kidney transplant data, warm ischaemia and complications were predictive of graft function at one year (P = 0.017, $\rho$ = 0.431 and P = 0.010, $\rho$ = -0.450 respectively). Among the fifteen models, three models with the highest coefficients of determination stand out. They were models 13, 14 and 15 (Table 7: P = 0.000, $r^2$ = 0.599, $r^2$adj = 0.549; P = 0.000, $r^2$ = 0.548, $r^2$adj = 0.497; P = 0.000, $r^2$ = 0.553, $r^2$adj = 0.517 respectively).

To put these findings in perspective, in many areas of the social and biological sciences, an $r^2$ of about 0.50 or 0.60 is considered high [41], and the number of 10 observations per predictor is required, i.e. 30 observations for the 3 predictors (eGFR-MDRD D6 or D21 or M3, NO D6 and age donor).

In addition, the value of the adjusted $r^2$ differs slightly from that of the $r^2$, which indicates the reliability of our models. The adjusted $r^2$ shows the interest of the cumulative effect of adding a variable to a model. In multiple linear regression, the addition of a variable to a model increases systematically $r^2$ without systematically increasing the value of the adjusted $r^2$.

The p-values of these three models are highly significant $P = 0.000$ and the application conditions are met (normality of residuals, homoscedasticity of residuals and absence of multicollinearity). Also, the p-values for the variable NO D6 were also significant in models 13, 14 and 15 ($P = 0.002$, $P = 0.011$, $P = 0.000$, respectively).

In addition, model 13, which represents the best model, can predict graft function at 6 months on the basis of quantification of creatininemia levels (allowing calculation of eGFR-MDRD) and nitric oxide levels in the blood on day 6. One unit of eGFR-MDRD D6 increases eGFR-MDRD M6 by 0.339 (coefficient of regression), similarly one unit of NO increases eGFR-MDRD M6 by 0.226 (coefficient of regression). The coefficients of the two serum biomarkers are relatively similar.

This study shows for the first time that NO measurement on day six post-transplant can be a predictive marker of eGFR at six months. Despite these encouraging results, there is a need for evaluation and validation in a larger cohort and by other centers. To our knowledge, only two studies have investigated the use of oxidative stress biomarkers for prediction of kidney graft function. In 2014, a study concluded that malondialdehyde level on day 7 might represent a useful predictor of one year serum creatinine [22]. The second was conducted in 2021 and highlighted the association of higher levels of free thiols at day 1 and day 5 with higher measured GFR at day 5 as well as with measured GFR at one year [23]. In addition, increased MDA levels on day 1 after kidney transplantation might be an early prognostic indicator of DGF and plasma levels of free thiols at 30 minutes and 90 minutes post-transplantation, were significantly higher among patients experiencing DGF [22, 23].

NO is a labile radical gas generated endogenously by a large number of cells throughout the body via three different NOS isoenzymes. Neuronal NOS (nNOS; also known as NOS1) and endothelial NOS (eNOS; also known as NOS3) are constitutively expressed, while inducible NOS (iNOS; also known as NOS2) is mainly associated with inflammatory conditions, which explains the duality between the beneficial and deleterious effects of NO [42]. NO is involved in the kidney's autoregulatory mechanisms, which aim to keep blood flow and GFR relatively constant despite variations in renal perfusion pressure over a wide range (80–180 mmHg). These mechanisms are essential for preventing barotrauma via the myogenic response, tubulo-glomerular feedback derived from the macula densa and their interactions, as well as making a substantial contribution to renal sodium and water management by inhibiting tubular sodium reabsorption along the nephron in response to the renin-angiotensin-aldosterone system [43]. L-Arginine, molecular oxygen, NADPH and tetrahydrobiopterin (BH4) are equally important substrates or cofactors that lead to the equimolar generation of NO and L-citrulline [43]. It is known that decreased bioavailability of NO or deficiency of NO production due to decreased availability of the substrate, L-arginine or an increased asymmetric dimethylarginine (ADMA), a potent inhibitor of endothelial NO synthase, causes development of cardiovascular diseases and chronic kidney diseases, and is highly associated with aging [44]. In an 11-year follow-up of a cohort of 1407 healthy participants of average age (58) and Northern European origin, the association of serum levels of endogenous NO inhibitors ADMA and symmetric dimethylarginine (SDMA), as well as the NO precursors (arginine, citrulline and ornithine) with a decline in mGFR and the development of CKD (mGFR < 60 ml/min per 1.73 m$^2$) were studied. Higher levels of SDMA were associated with a slower annual decline in GFR, while higher levels of citrulline and ornithine were associated with an accelerated decline in GFR. Higher levels of citrulline were associated with the development of chronic kidney disease

[45]. In a small cohort study of 25 patients with different stages of CKD and 25 healthy subjects, NO showed a significant positive correlation with eGFR (r = 0.476, P = 0.016) in patients with stage 3 and 4 CKD; and citrulline showed a significant positive correlation with creatinine in patients with stage 1 and 2 CKD (r = 0.415, P = 0.044) [46]. In addition, the authors of a study investigating the effects of chronic dietary supplementation with L-arginine on kidney aging do not recommend long-term dietary supplementation with L-arginine, particularly among the elderly. In fact, such supplementation accelerates the functional decline of the kidneys and vascular system as people age. The study was conducted on young mice (4 months old) and elderly mice (18–24 months old), given either a standard diet containing 0.65% L-arginine or a diet supplemented with 2.46% L-arginine for 16 weeks. L-arginine supplementation further increased age-associated albuminuria and mortality, particularly in females, and was accompanied by elevated levels of renal arginase-II (Arg-II). L-arginine supplementation increases ROS and decreases NO production in the aortas of aged mice [47]. On the other hand, a prospective study done on 50 recipients of renal allografts revealed a significant increase of serum nitrate and episodes of acute rejection compared with other causes of renal dysfunction (delayed graft function, urinary tract infection and tacrolimus toxicity). The authors suggest that NO is a useful marker to aid in the diagnosis of rejection [35]. Serum NO level is also a prognostic parameter for chronic rejection, as shown by the authors of a study who suggested that continuously elevated serum NO levels could predict graft loss 6 to 12 months earlier. Threshold values of 150 mmol/L nitric oxide products and 3 mg/dL serum creatinine were found to be positively predictive of graft loss within one year [48].

As for the influence of donor's parameters, a number of studies have been carried out. A study conducted in 2018 on donor-recipient function correlation concluded that, excluding unpredictable complications in the post-transplant period, the donor's pre-donation eGFR, eGFR in donors at hospital discharge and age were the best predictors of recipient and donor eGFR after one year and can be used as a tool to manage expectations for the post-transplant period [49]. In 2020, a retrospective study of a cohort of 290 pairs of donors and recipients who had undergone a kidney transplant from living donors revealed by means of univariate linear regression analysis, that the donor kidney weight/recipient body weight, donor kidney weight/recipient body surface area, donor kidney weight/recipient body mass index, donor kidney volume/recipient body weight, donor kidney volume/recipient body surface area, donor kidney volume/recipient body mass index, and donor body weight/recipient body weight were significantly correlated with eGFR and serum creatinine in recipients within two years of transplantation. In multivariate linear regression analysis, the donor kidney weight/recipient body weight ratio and donor age were significantly correlated with eGFR at 6, 12, 18 and 24 months post-transplant, with the donor kidney weight/recipient body weight ratio performing better in predicting good renal allograft function at 12 months post-transplant [50]. Moreover, a model for predicting early graft function one month after kidney transplantation from living donors has been developed and validated in 2022. The model includes the ratio of cortex weight to recipient weight and the donor's eGFR as preoperative predictors [49, 51]. Cortical volume predicts graft function at one year in living donors and the amount of interstitial fibrosis predicts graft function at one year in recipients. This is the conclusion of a study carried out in 2023 on 49 living kidney donors and 51 recipients, in which the associations between GFR at one-year post-transplant in donors and recipients with cortical volume and histomorphometric parameters (total number of glomeruli, glomerular volume, glomerular sclerosis, renal fibrosis and arteriolar dimensions) has been explored [52]. In 2019, a study developed and validated a new model for predicting graft function using pre-operative marginal factors in living donor kidney transplantation, using four preoperative variables as predictors, namely donor age, donor eGFR, donor hypertension and donor-recipient body weight ratio [53].

The results of all these studies agree that the donor's age is a predictor of graft function. Twelve-month eGFR is a strong predictor of long-term graft failure when taken into account for clinical events occurring from discharge to one year. These findings may improve patient management and clinical evaluation of further interventions [3]. In our study, the best model for predicting graft function one year after transplantation (P = 0.000, $r^2$ = 0.690, $r^2$adj = 0.655) included the following parameters: complication, warm ischaemia and recipient GFR estimated at six months, the latter representing the best predictor among all the MDRD-eGFR variables. Early prediction of GFR one year after transplantation was carried out in a large multi-centre cohort of 376 patients. A panel of biomarkers including gene expression, cytokine, metabolomic and antibody reactivity profiles, in the pre- and early post-transplant period were analysed. The pre-transplant data yielded a Pearson correlation coefficient of r = 0.39 between measured and predicted GFR at one year. Two weeks after transplantation, the correlation increased to r = 0.63, and at three months, to r = 0.76. The authors point out that eGFR showed remarkable stability, achieving similar results on its own to the full subset of clinical markers. They investigated whether demographic factors added value to the prediction of 1-year GFR compared to eGFR alone. Multiparametric regression revealed that recipient age, donor age and donor CMV serostatus at 2 weeks were independently associated with eGFR at 1 year (recipient age: P = 0.045; donor age: P<0.001; donor CMV serostatus: P = 0.004; eGFR at 2 weeks: P<0.001). Similarly, the association between donor age and BMI at 3 months with eGFR at 1 year was independent of eGFR at 3 months (donor age: P<0.001; BMI: P = 0.008; eGFR at 3 months: P<0.001). They concluded that these variables are considered to be prognostic factors [54]. On the other hand, the performance of equations for estimating GFR based on serum creatinine was compared with that of GFR measured during 20 years of longitudinal follow-up in a single-study centre of 417 transplant patients. All eGFR equations showed similar trends toward measured GFR, following a decline in GFR, no significant differences were observed in the individual changes (slopes) of measured GFR or estimated GFR in predicting graft loss in the coming months or years. However, the percentage of transplant patients with a >30% decline in GFR in the last period before graft loss was significantly lower for the estimated GFR than for the measured GFR, with discordant measured GFR results in ~25% of cases [55]. In 2023, a race-free estimated glomerular filtration rate equation specific to kidney transplant recipients and based on creatinine measurement, was developed and validated in a multiple large international cohorts of recipients from Europe, United States, South America, Canada, Asia, Africa and Oceania. It showed significantly improved performance compared to the race-free 2021 CKD-EPI equation (developed in individuals with native kidneys) and performed well in the external validation cohorts [56]. The P30 values (P30 being the proportion of eGFR within 30% of measured GFR) ranged from 73.0% to 91.3% [56].

In conclusion, the results of our study suggest that nitric oxide quantification at day six will be useful in predicting graft function at six months in living donor kidney transplants. Furthermore, among the demographic, clinical, induction/primary immunossupressive treatment, complications, rejection, and transplant-related parameters of the living donor-recipient pair, only donor age is predictive of graft function at six months. The combination of factors, complications, warm ischaemia and MDRD-eGFR at six months offers a better prediction model of glomerular filtration rate one year after transplantation.

## Acknowledgments

The authors are extremely appreciative for the commitment and crucial help of the nursing staff for sample collection. The authors are extremely grateful to Dr. Safia Zenia statistician for

her help in statistical analysis. The authors are deeply grateful to Mr. Kamel Babouche for his linguistic corrections.

## Author Contributions

**Conceptualization:** Djamila Izemrane, Ali Benziane, Mohamed Makrelouf, Nacim Hamdis, Samia Hadj Rabia, Sofiane Boudjellaba, Ahsene Baz, Djamila Benaziza.

**Data curation:** Djamila Izemrane, Sofiane Boudjellaba.

**Formal analysis:** Djamila Izemrane, Sofiane Boudjellaba.

**Funding acquisition:** Djamila Izemrane, Mohamed Makrelouf, Nacim Hamdis, Ahsene Baz, Djamila Benaziza.

**Investigation:** Djamila Izemrane.

**Methodology:** Djamila Izemrane, Ali Benziane, Mohamed Makrelouf, Nacim Hamdis, Samia Hadj Rabia, Sofiane Boudjellaba, Ahsene Baz, Djamila Benaziza.

**Project administration:** Ahsene Baz, Djamila Benaziza.

**Resources:** Ali Benziane, Mohamed Makrelouf, Nacim Hamdis, Samia Hadj Rabia, Ahsene Baz.

**Software:** Djamila Izemrane, Sofiane Boudjellaba.

**Supervision:** Ali Benziane, Ahsene Baz.

**Validation:** Djamila Izemrane, Ali Benziane, Ahsene Baz.

**Visualization:** Djamila Izemrane, Ali Benziane, Ahsene Baz.

**Writing – original draft:** Djamila Izemrane, Ali Benziane, Ahsene Baz.

**Writing – review & editing:** Djamila Izemrane, Ali Benziane, Mohamed Makrelouf, Nacim Hamdis, Samia Hadj Rabia, Sofiane Boudjellaba, Ahsene Baz, Djamila Benaziza.

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
