## [Decision Letter · Decision Letter 0]

2 Nov 2023

PONE-D-23-24980Living donors kidney transplantation and oxidative stress: nitric oxide as a predictive marker of graft functionPLOS ONE

Dear Dr. IZEMRANE,

Thank you for submitting your manuscript to PLOS ONE. After careful consideration, we feel that it has merit but does not fully meet PLOS ONE’s publication criteria as it currently stands. Therefore, we invite you to submit a revised version of the manuscript that addresses the points raised during the review process.

ACADEMIC EDITOR:  Dear authors, thank you for submitting your manuscript to PLOS One. The reviewers have commented on your manuscript, and I would suggest that you address all of the comments, especially the language issues.

We look forward to receiving your revised manuscript.

Kind regards,

Belal Nedal Sabbah

Academic Editor

PLOS ONE

“This article was written as part of a doctoral thesis.  The funding comes from state institutions, namely the laboratory of biology and animal physiology of Higher Normal School of Kouba, Algiers, Algeria and the central biology laboratory of Lamine Debaghine University Hospital, Bab El Oued, Algiers, Algeria and personnal funding from Djamila Izemrane and  Nacim Hamdis.”

Reviewers' comments:

Reviewer's Responses to Questions

**Comments to the Author**

1. Is the manuscript technically sound, and do the data support the conclusions?

Reviewer #1: No

Reviewer #2: Yes

2. Has the statistical analysis been performed appropriately and rigorously? 

Reviewer #1: No

Reviewer #2: Yes

3. Have the authors made all data underlying the findings in their manuscript fully available?

Reviewer #1: Yes

Reviewer #2: No

4. Is the manuscript presented in an intelligible fashion and written in standard English?

Reviewer #1: No

Reviewer #2: Yes

5. Review Comments to the Author

Reviewer #1: The authors have evaluated the relationship between blood levels of oxidative stress markers and eGFR after living donor kidney transplantation.

The topic was interesting and meaningful.

However, poor English made it difficult to understand.

Probably due to the small sample size, multivariate analysis has shown that oxidative stress markers cannot predict eGFR. It is inappropriate to describe as if they have predictive ability despite these findings.

In the large number of multiple comparisons made in the first half of the study, it is not surprising that some of them were falsely “statistically significant” due to alpha errors.

What makes it different and novel from similar reports already published?

I think it is not suitable for publication as is.

Reviewer #2: Dear authors,

This is an excellent idea and well written manuscript. However some points:

1- the manuscript is too lengthy and it should be rewritten concisely.

2- Some of the tables are very complicated and need to be edited.

3- More detailed information in methods, regarding donor specification, surgeons and surgical techniques.

6. PLOS authors have the option to publish the peer review history of their article (what does this mean?). If published, this will include your full peer review and any attached files.

Reviewer #1: No

Reviewer #2: **Yes: **Seyed Reza Yahyazadeh

---

## [Author Response · Author response to Decision Letter 0]

14 Dec 2023

Dear Academic Editors and Peer Reviewers

 We thank you so much for the time you have devoted to reading and understanding this manuscript, as well as for your valuable comments and relevant remarks. 

 Further to your observations about the language of the article, I submitted the latter to Mr. Kamel Babouche, who holds a long experience in English-language teaching and training as a:

- Former high-school teacher of English, (1978-1990).

- Former English-language teacher trainer at a teaching school, (1990-1996).

- Former general inspector for English, (1996-2014).

- Former consultant at British Council Algeria, (2014-2018).

 Mr. Babouche read through the article to see how best he could improve it in terms of language quality. Though he admitted there is much redundancy, his view is that the nature of the text, as well as the jargon in relation to the discipline allow for a certain amount of redundancy as regards the syntax. He reckons it is difficult to bring syntactic changes without running the risk of altering the meaning and comprehension of the article, and so limited himself to correcting a few grammatical and lexical mistakes. All mistakes are corrected in green in the manuscript titled "Revised Manuscript with Track Changes".

 I also wish to draw your attention to the fact that English language -as a tool of teaching and studying at university- has been introduced only recently in the Algerian universities and is gradually replacing French. This means that we have to make more effort to reach good standards of English and we are doing our best for that purpose. Therefore I hope to rely on your understanding and indulgence as regards the linguistic quality of the article.

 In addition, we have responded to the 6 points required by PLOS ONE, as well as to the reviewers' comments.

PLOS ONE requirements 

1- The formatting of the unmarked version complies with the recommendations of the link https://journals.plos.org/plosone/s/file?id=wjVg/PLOSOne_formatting_sample_main_body.pdf

2- Response to Emily Chenette, Editor-in-Chief of PLOS ONE, and Iain Hrynaszkiewicz, Director of Open Research Solutions at PLOS: We created the doi of our raw data according to the registrations on the https://plos.org/dryad-data/ website.

3- We have specified in the cover letter that funding from the two laboratories (Animal Biology and Physiology laboratory at the Ecole Normale Supérieure in Kouba, Algiers, Algeria and the Central Biology Laboratory at the CHU Lamine Debaghine, Bab El Oued, Algiers, Algeria) is used to finance our analyses.

4- We have mentioned in the cover letter and in green in the statistics section of the manuscript titled "Revised Manuscript with Track Changes" that our raw data are available to all scientific readers via https://datadryad.org/stash/share/2amXBXayUq0oPVI2xg_IoF9_04VM3u7AFdfjfNtymFE , lines 705 and 706.

5- The words "data not shown" have been removed from the text and appear in red in the "Results section/ Construction of prediction models of graft function" of the manuscript titled "Revised Manuscript with Track Changes", line 323.

6- We have included our full ethics statement in the "Methods" section of the manuscript titled "Revised Manuscript with Track Changes", lines 138,139 and 140.

Response to reviewer 1 Anonymous

1- As mentioned above, grammatical and lexical errors have been corrected by Mr. Kamel Babouche.

2- We would like to make a small clarification: we claim that the quantification of nitric oxide at day 6 in association of MDRD-eGFR D6, or MDRD-eGFR D21, or MDRD-eGFR M3 and/or donor age makes it possible to predict the glomerular filtration rate at 6 months, and consequently, the functioning of the kidney transplant in patients transplanted from living donors (see models 13, 14 and 15 table 6b) (Respectively, P= 0.000, r2 = 0.599, r2adj= 0.549 ; P= 0.000, r2= 0.548, r2adj = 0.497 ; P= 0.000, r2= 0.553, r2adj = 0.517). The other three biomarkers (malondialdehyde, myeloperoxidase and gluthathione s transferase) were not predictive in this study. 

In many areas of the social and biological sciences, an r2 of about 0.50 or 0.60 is considered high (R.D. Cook and S. Weisberg (1999), Applied Regression Including Computing and Graphics, Wiley, p. 281), and the number of 10 observations per predictor is required, i.e. 30 observations for the 3 predictor variables (eGFR-MDRD D6 or D21 or M3, NO D6, age donor), so my sample of 32 patients is correct.

In addition, the value of the adjusted r2 differs slightly from that of the r2, which indicates the reliability of our models. The adjusted r2 shows the interest of the cumulative effect of adding a variable to a model. In multiple linear regression, the addition of a variable to a model increases systematically r2 without systematically increasing the value of the adjusted r2. 

On another note, the P values of these three models are highly significant P=0.000 and the application conditions are met (normality of residuals, homoscedasticity of residuals and absence of multicollinearity). Also, the P values for the variable NO D6 were also significant in models 13, 14 and 15 (P= 0.002, P=0.011, P=0.000, respectively).

In addition, in model 13, which represents the best model, we can predict graft function at 6 months on the basis of quantification of creatininemia levels (allowing calculation of eGFR-MDRD) and nitric oxide levels in the blood on day 6. 1 unit of eGFR-MDRD D6 increases eGFR-MDRD M6 by 0.339 (coefficient of regression), similarly 1 unit of NO increases eGFR-MDRD M6 by 0.226 (coefficient of regression). The coefficients of the two serum biomarkers are relatively similar. 

All these clarifications are provided from line 550 to line 565 and colored in green, in the document entitled "Revised Manuscript with Track Changes".

3- The first part of the results has been pruned to target the objectives. It is clear that some of the P values may be debatable.

4- The originality of this article is that, for the first time, modified MDRD equations have been developed to predict renal graft function at 6 months. These equations take into account two serum biomarkers nitric oxide (a biomarker associated with GFR according to references (36, 50, 51 and 52) and creatininemia, in addition to donor age. Several studies on donor parameters agree that donor age predicts graft function (references 55, 56, 57, 58 and 59).

However, this work deserves to be completed by validating these equations on a large cohort of kidney transplant recipients from living donors, or even testing these equations on a cadaveric cohort. 

Response to reviewer 2 Mr. Seyed Reza Yahazadeh

1- Indeed, some readers found the manuscript long. Some parts of the text have been trimmed to make the results easier. All the parts concerned are marked in red (Results and discussion) in the document entitled "Revised Manuscript with Track Changes", from line 268 to line 310 and from line 496 to line 500. Also, two of the bibliographic references, namely 46 and 47, are no longer listed.

2- Some readers also found the tables difficult to read. A change has been made to the layout of tables 6a and 6b. The variables introduced and significant for each of the models are listed at the bottom of the table and not in the table itself. Table 7 has been completely modified to make the equations of the 15 models used to predict graft function at 6 months easier to read. These changes are marked in green in the document entitled "Revised Manuscript with Track Changes".

3- We have added a few details concerning surgery and donors as requested in the "Methods" section, namely ABO-compatible transplantation, spouses as unrelated living donors, uretero-vesical anastomosis in most cases, Celcior preservation fluid used and left kidney transplant, lines 135, 140 and 141. These details are marked in green in the document named "Revised Manuscript with Track Changes".

Sincerely yours,

Djamila IZEMRANE, Ali BENZIANE and Ahcène BAZ on behalf of the authors.

Corresponding author: Djamila IZEMRANE. 

- Laboratory of Biology and Animal Physiology, Higher Normal School, Kouba, Algiers, Algeria, 16308.

- National Higher Veterinary School, Issad Abbes, Oued Smar, Algiers, Algeria, 16059.

Email: d.izemrane@ensv.dz. izemranedjamila@gmail.com. Phone number: +213 669 20 82 01 / +213 772 21 67 32.

---

## [Decision Letter · Decision Letter 1]

4 Jan 2024

PONE-D-23-24980R1

Living donors kidney transplantation and oxidative stress: nitric oxide as a predictive marker of graft function

PLOS ONE

Dear Dr. IZEMRANE,

Thank you for submitting your manuscript to PLOS ONE. After careful consideration, we have decided that your manuscript does not meet our criteria for publication and must therefore be rejected.

I am sorry that we cannot be more positive on this occasion, but hope that you appreciate the reasons for this decision.

Kind regards,

Belal Nedal Sabbah

Academic Editor

PLOS ONE

Additional Editor Comments:

**The reviewers have indicated that the manuscript in its current state is not suitable for publication. Please find the comments below for your own reference.**

Reviewers' comments:

Reviewer's Responses to Questions

**Comments to the Author**

1. If the authors have adequately addressed your comments raised in a previous round of review and you feel that this manuscript is now acceptable for publication, you may indicate that here to bypass the “Comments to the Author” section, enter your conflict of interest statement in the “Confidential to Editor” section, and submit your "Accept" recommendation.

Reviewer #1: (No Response)

2. Is the manuscript technically sound, and do the data support the conclusions?

Reviewer #1: No

3. Has the statistical analysis been performed appropriately and rigorously? 

Reviewer #1: No

4. Have the authors made all data underlying the findings in their manuscript fully available?

Reviewer #1: Yes

5. Is the manuscript presented in an intelligible fashion and written in standard English?

Reviewer #1: No

6. Review Comments to the Author

Reviewer #1: The authors have submitted a revised manuscript, but it isn't easy to understand how the paper has changed. It would be better to include the reviewer's remarks and a point-by-point response to each of them in the response letter.

There are still typographical errors in the revised manuscript and the problems I pointed out regarding multiple comparisons have not been resolved. There are numerous multivariable stepwise linear regression analyses in which only a few variables are selected for inclusion. Some of these analyses will likely produce good results by chance. In addition, the choice of variables seems somewhat arbitrary and raises the question of why all the information in Table 4 was not included in the analysis.

7. PLOS authors have the option to publish the peer review history of their article (what does this mean?). If published, this will include your full peer review and any attached files.

Reviewer #1: No

- - - - -

---

## [Author Response · Author response to Decision Letter 1]

24 Jan 2024

We have responded to the 6 points required by PLOS ONE, as well as to the reviewers' comments.

PLOS ONE requirements 

1- The formatting of the unmarked version complies with the recommendations of the link https://journals.plos.org/plosone/s/file?id=wjVg/PLOSOne_formatting_sample_main_body.pdf

2- Response to Emily Chenette, Editor-in-Chief of PLOS ONE, and Iain Hrynaszkiewicz, Director of Open Research Solutions at PLOS: We created the doi of our raw data according to the registrations on the https://plos.org/dryad-data/ website.

3- We have specified in the cover letter that funding from the two laboratories (Animal Biology and Physiology laboratory at the Ecole Normale Supérieure in Kouba, Algiers, Algeria and the Central Biology Laboratory at the CHU Lamine Debaghine, Bab El Oued, Algiers, Algeria) is used to finance our analyses.

4- We have mentioned in the cover letter and in green in the statistics section of the manuscript titled "Revised Manuscript with Track Changes" that our raw data are available to all scientific readers via https://datadryad.org/stash/share/2amXBXayUq0oPVI2xg_IoF9_04VM3u7AFdfjfNtymFE , lines 705 and 706.

5- The words "data not shown" have been removed from the text and appear in red in the "Results section/ Construction of prediction models of graft function" of the manuscript titled "Revised Manuscript with Track Changes", line 323.

6- We have included our full ethics statement in the "Methods" section of the manuscript titled "Revised Manuscript with Track Changes", lines 138,139 and 140.

Response to reviewer 1 Anonymous

1- As mentioned above, grammatical and lexical errors have been corrected by Mr. Kamel Babouche.

2- We would like to make a small clarification: we claim that the quantification of nitric oxide at day 6 in association of MDRD-eGFR D6, or MDRD-eGFR D21, or MDRD-eGFR M3 and/or donor age makes it possible to predict the glomerular filtration rate at 6 months, and consequently, the functioning of the kidney transplant in patients transplanted from living donors (see models 13, 14 and 15 table 6b) (Respectively, P= 0.000, r2 = 0.599, r2adj= 0.549 ; P= 0.000, r2= 0.548, r2adj = 0.497 ; P= 0.000, r2= 0.553, r2adj = 0.517). The other three biomarkers (malondialdehyde, myeloperoxidase and gluthathione s transferase) were not predictive in this study. 

In many areas of the social and biological sciences, an r2 of about 0.50 or 0.60 is considered high (R.D. Cook and S. Weisberg (1999), Applied Regression Including Computing and Graphics, Wiley, p. 281), and the number of 10 observations per predictor is required, i.e. 30 observations for the 3 predictor variables (eGFR-MDRD D6 or D21 or M3, NO D6, age donor), so my sample of 32 patients is correct.

In addition, the value of the adjusted r2 differs slightly from that of the r2, which indicates the reliability of our models. The adjusted r2 shows the interest of the cumulative effect of adding a variable to a model. In multiple linear regression, the addition of a variable to a model increases systematically r2 without systematically increasing the value of the adjusted r2. 

On another note, the P values of these three models are highly significant P=0.000 and the application conditions are met (normality of residuals, homoscedasticity of residuals and absence of multicollinearity). Also, the P values for the variable NO D6 were also significant in models 13, 14 and 15 (P= 0.002, P=0.011, P=0.000, respectively).

In addition, in model 13, which represents the best model, we can predict graft function at 6 months on the basis of quantification of creatininemia levels (allowing calculation of eGFR-MDRD) and nitric oxide levels in the blood on day 6. 1 unit of eGFR-MDRD D6 increases eGFR-MDRD M6 by 0.339 (coefficient of regression), similarly 1 unit of NO increases eGFR-MDRD M6 by 0.226 (coefficient of regression). The coefficients of the two serum biomarkers are relatively similar. 

All these clarifications are provided from line 550 to line 565 and colored in green, in the document entitled "Revised Manuscript with Track Changes".

3- The first part of the results, from line 270 to line 310 and colored in red in the document entitled "Revised Manuscript with Track Changes" has been pruned to target the objectives. It is clear that some of the P values may be debatable. 

4- The originality of this article is that, for the first time, modified MDRD equations have been developed to predict renal graft function at 6 months. These equations take into account two serum biomarkers nitric oxide (a biomarker associated with GFR according to references (36, 50, 51 and 52) and creatininemia, in addition to donor age. Several studies on donor parameters agree that donor age predicts graft function (references 55, 56, 57, 58 and 59).

However, this work deserves to be completed by validating these equations on a large cohort of kidney transplant recipients from living donors, or even testing these equations on a cadaveric cohort. 

Response to reviewer 2 Mr. Seyed Reza Yahazadeh

1- Indeed, some readers found the manuscript long. Some parts of the text have been trimmed to make the results easier. All the parts concerned are marked in red (Results and discussion) in the document entitled "Revised Manuscript with Track Changes", from line 268 to line 310 and from line 496 to line 500. Also, two of the bibliographic references, namely 46 and 47, are no longer listed.

2- Some readers also found the tables difficult to read. A change has been made to the layout of tables 6a and 6b. The variables introduced and significant for each of the models are listed at the bottom of the table and not in the table itself. Table 7 has been completely modified to make the equations of the 15 models used to predict graft function at 6 months easier to read. These changes are marked in green in the document entitled "Revised Manuscript with Track Changes".

3- We have added a few details concerning surgery and donors as requested in the "Methods" section, namely ABO-compatible transplantation, spouses as unrelated living donors, uretero-vesical anastomosis in most cases, Celcior preservation fluid used and left kidney transplant, lines 135, 140 and 141. These details are marked in green in the document named "Revised Manuscript with Track Changes".

---

## [Decision Letter · Decision Letter 2]

18 Apr 2024

PONE-D-23-24980R2

Living donors kidney transplantation and oxidative stress: nitric oxide as a predictive marker of graft function

PLOS ONE

Dear Dr. IZEMRANE,

Thank you for submitting your manuscript to PLOS ONE. After careful consideration, we feel that it has merit but does not fully meet PLOS ONE’s publication criteria as it currently stands. Therefore, we invite you to submit a revised version of the manuscript that addresses the points raised during the review process.

We look forward to receiving your revised manuscript.

Kind regards,

John Richard Lee, M.D.

Academic Editor

PLOS ONE

Journal Requirements:

Additional Editor Comments (if provided):

Reviewers' comments:

Reviewer's Responses to Questions

**Comments to the Author**

1. If the authors have adequately addressed your comments raised in a previous round of review and you feel that this manuscript is now acceptable for publication, you may indicate that here to bypass the “Comments to the Author” section, enter your conflict of interest statement in the “Confidential to Editor” section, and submit your "Accept" recommendation.

Reviewer #3: (No Response)

Reviewer #4: All comments have been addressed

Reviewer #5: (No Response)

2. Is the manuscript technically sound, and do the data support the conclusions?

Reviewer #3: No

Reviewer #4: Yes

Reviewer #5: Yes

3. Has the statistical analysis been performed appropriately and rigorously? 

Reviewer #3: Yes

Reviewer #4: Yes

Reviewer #5: Yes

4. Have the authors made all data underlying the findings in their manuscript fully available?

Reviewer #3: Yes

Reviewer #4: Yes

Reviewer #5: Yes

5. Is the manuscript presented in an intelligible fashion and written in standard English?

Reviewer #3: Yes

Reviewer #4: Yes

Reviewer #5: Yes

6. Review Comments to the Author

Reviewer #3: While the authors have attempted to generate a model to predict renal allograft graft function at one year using markers of oxidative stress in a living kidney donation model, there is no validation cohort to test the model.

Reviewer #4: The previous reviewers comments have been reviewed and edits to the current manuscript as well. The paper is improved in its current form, however, I would consider adjusting the following:

1. The paper continues to be quite lengthy. The introduction and discussion should be limited to to only background and details that relate to findings of the paper. For example in the first paragraph the discussion about the MDRD and measuring of eGFR does not need to be as detailed as it is and the authors can focus only on the limitations of the current testing which is why they proceeded with their study. Similarly in the discussion, the detailed discussion of the biomarkers studies in the paper and the evidence for their usage is likely not needed and the focus of the discussion should be on the specific findings of the paper and the review of other references that support their findings should be more succinct

2. Were there different findings in terms of the biomarkers in terms of DGF? Were levels of sNO or the other biomarkers higher in the patients that had DGF as compared with those whose grafts functioned immediately?

Reviewer #5: The paper is an interesting examination however there a multiple typos and grammatical errors throughout the manuscript. Ones that I saw were:

Line 58: recognized; Line 86: oxygen; Line 105: hemodialysis; Line 110: energy demanding, mitochondria; Line 117: In 2014, a; Line 119: measured; Line 123: living donors. It also; Line 124: transplantation as well; Line 129: October 2017 and November; Line 131: living donors.; Line 133: study. All; Line 134: The left kidney was transplanted and the type; Line 169: A volume of 100 ul, 2900 ul; Line 222: therapy consisting of calcineurin; Line 226: Table 1; Line 433 correlation results, which; Line 434: at 6M, shows; Line 501: aging; Line 576: race-free; Table 1: Rejection, Brother, Cousins; Table 5: Rejection

Regarding formatting their data, I felt it was confusing at times. For example, in Table 1, the authors displays Age (yr) then the data is 39.8 (10.6). The convention is that what is in the parenthesis matches in the data. However regarding the data I believe the authors wanted to represent it as 39.8±10.6 showing the standard deviation of the number. If that is so, I would covert all the data were appropriate to the ± format.

In the methods; it would be useful to describe the centers' calcineurin inhibitor trough goals over the first year. Moreover, this maybe beyond the scope of their analysis, but would they able to see if there is a correlation between average calcineurin levels and GFR and average calcineurin levels and the the levels of their biomarkers. For example, higher levels of calcineurin inhibitors would constrict the afferent arterioles and perhaps increase NO production in the the efferent arterioles in the kidney. Maybe this effect is not only happening in the glomerulus but also systemically and can be can be seen in the blood. If they can perform that analysis perhaps they can comment on the possible effects of calcineurin inhibitors on their biomarkers.

Regarding Table 1; since the authors are using the MDRD equation it would be interesting in the demographic data to see the percentage of patients that fall into the African versus non-African calculations to get more information regarding the examined population.

If possible, Table 3 should be converted into a figure showing the dot plots with correlations of the 3 significant findings MDRD 6M/MPO 1M, MDRD 6M/NO D6 and MDRD 6M/NO D21 which would highlight their findings. The table can be moved to a supplemental section if readers want to view it.

7. PLOS authors have the option to publish the peer review history of their article (what does this mean?). If published, this will include your full peer review and any attached files.

Reviewer #3: No

Reviewer #4: No

Reviewer #5: No

---

## [Author Response · Author response to Decision Letter 2]

9 May 2024

Algiers, May 9th 2024.

Dear Academic Editor and Peer Reviewers

 We thank you so much for the time you have devoted to reading and understanding this manuscript. We would also like to express our gratitude for the interest you have shown in our work, which is reflected in your pertinent comments and recommendations for improving the manuscript.

 We have responded to the 2 points required by PLOS ONE, as well as to the reviewers' comments.

PLOS ONE requirements 

1- Please note that the content of our financial disclosure was changed in the cover letter. We have removed the reference to personal contributions by Djamila IZEMRANE and Nacim HAMDIS.

2- As far as depositing our laboratory protocols in protocols.io is concerned, we feel that there is no need to make an additional contribution, given that these protocols are referenced (see methods section) and that the analysis methods are common methods with few modifications.

Response to reviewers

Reviewer#3

Despite these encouraging results, there is a need for evaluation and validation in a larger cohort and by other centers. This is mentioned in the lines 474, 475 of the document entitled "Revised Manuscript with Track Changes". 

Cohorts for the creation and validation of predictive models are generated using the 20-80 or 30-70 method. Given that the number of patients transplanted to date at the University Hospital of Beb El Oued, with a follow-up of one year after the last patient in our cohort, is 58 patients. We cannot therefore validate a posteriori our predictive model of graft function at one year. 

In addition, it is impossible to obtain D6 blood samples from these patients in order to repeat the NO assays for validation of the models predicting graft function at six months.

Reviewer#4 

1- We have followed your recommendations by deleting some paragraphs from the introduction (lines 69-73, lines 79-83, lines 99-101 and lines 103-106) and from the discussion (lines 386,387, lines 389-395 and lines 400-407). All of these paragraphs are highlighted in red in the document entitled "Revised Manuscript with Track Changes". This has resulted in the deletion of references (10, 11, 22, 41, 42 and 43) and a change in numbering from 12th reference.

2- For patients with DGF compared to patients with prompt graft function, the activity of sMPO was higher for patients with DGF. The significant differences were observed at D6 and D8 (respectively, 112.04±44.32 vs. 49.89±39.49 U/L, P=0.023 and 126.81±59.29 vs. 47.48±30.72 U/L, P=0.01). 

In addition, levels of sNO at D8 posttransplantation (99.86±44.22 vs. 45.77±40.0 µmole/L, P=0.027), higher in favour of the DGF group. 

Reviewer#5 

1- Typos and grammatical errors in the docuement entitled "Revised Manuscript with Track Changes" have been corrected : Line 59: recognized; Line 87: oxygen; Line 106: hemodialysis; Line 111: energy demanding, mitochondria; Line 118: In 2014, a; Line 120: measured; Line 124: living donors. It also; Line 125: transplantation as well; Line 130: October 2017 and November; Line 132: living donors.; Line 134: study. All; Line 135: The left kidney was transplanted and the type; Line 170: A volume of 100 ul, 2900 ul; Line 223: therapy consisting of calcineurin; Line 227: Table 1; Line 434 correlation results, which; Line 435: at 6M, shows; Line 502: aging; Line 577: race-free; Table 1: Rejection, Brother, Cousins; Table 5: Rejection. All these corrections are hilighted in green in the document entiteled "Revised Manuscript with Track Changes"

2- The data format has been corrected and hilighted in green, in tables 1 and 2, as well as in the paragraphs mentioning the t-test results, from line 246 to line 251 and from line 256 to line 259 in the document entitled "Revised Manuscript with Track Changes"

3- We have clarified the immunosuppressive therapeutic strategy of our transplant centre in the method section of the manuscript entitled "Revised Manuscript with Track Changes", from line 225 to line 232

4- The results of the bivariate correlations between mean and individual NO values and mean and individual calcineurin inhibitor values did not reveal any significance, on any of the follow-up days. The same was true for the correlations between mean and individual glomerular filtration rate values and mean and individual calcineurin inhibitor values. Similarly, we mentioned in Table 5 that there is no correlation between glomerular filtration rate and the choice of calcineurin inhibitor.

5- The MDRD equation used is for patients of non-African origin, see table 1.

6- The authors prefer to keep table 3, which expresses the values of the Pearson and Spearman coefficients "r" and "ρ", rather than the scatterplot, which shows the shape or trend of the correlation, especially for a small sample size with a wide dispersion of values in relation to the mean. The significant values of these correlations are highlighted in grey in table 3.

Sincerely yours,

Djamila IZEMRANE, Ali BENZIANE and Ahcène BAZ on behalf of the authors.

Corresponding author: Djamila IZEMRANE. 

- Laboratory of Biology and Animal Physiology, Higher Normal School, Kouba, Algiers, Algeria, 16308.

- National Higher Veterinary School, Issad Abbes, Oued Smar, Algiers, Algeria, 16059.

Email: d.izemrane@ensv.dz. izemranedjamila@gmail.com. Phone number: +213 669 20 82 01

---

## [Decision Letter · Decision Letter 3]

26 Jun 2024

PONE-D-23-24980R3Living donors kidney transplantation and oxidative stress: nitric oxide as a predictive marker of graft functionPLOS ONE

Dear Dr. IZEMRANE,

Thank you for submitting your manuscript to PLOS ONE. After careful consideration, we feel that it has merit but does not fully meet PLOS ONE’s publication criteria as it currently stands. Therefore, we invite you to submit a revised version of the manuscript that addresses the points raised during the review process.

We look forward to receiving your revised manuscript.

Kind regards,

John Richard Lee, M.D.

Academic Editor

PLOS ONE

Journal Requirements:

Reviewers' comments:

Reviewer's Responses to Questions

**Comments to the Author**

1. If the authors have adequately addressed your comments raised in a previous round of review and you feel that this manuscript is now acceptable for publication, you may indicate that here to bypass the “Comments to the Author” section, enter your conflict of interest statement in the “Confidential to Editor” section, and submit your "Accept" recommendation.

Reviewer #3: All comments have been addressed

Reviewer #5: (No Response)

2. Is the manuscript technically sound, and do the data support the conclusions?

Reviewer #3: Yes

Reviewer #5: Yes

3. Has the statistical analysis been performed appropriately and rigorously? 

Reviewer #3: Yes

Reviewer #5: Yes

4. Have the authors made all data underlying the findings in their manuscript fully available?

Reviewer #3: Yes

Reviewer #5: Yes

5. Is the manuscript presented in an intelligible fashion and written in standard English?

Reviewer #3: Yes

Reviewer #5: Yes

6. Review Comments to the Author

Reviewer #3: Djamila et al present a study looking into using nitric oxide levels in recipients from living kidney donors as one of the predictors of graft function. Thank you for the revising the manuscript. I have no further critiques

Reviewer #5: Based on the most recent revisions I have no new analysis for the author. It is mostly gramatical and formating issues. Comments will have line numbers before them from the most recent revision:

62- kidney size (remove renal), 94- endothelial cell dysfunction, 125- remove "and ;"; remove "to", 126- post-transplantation, 142- blood samples were, 154- using the thiobarbituric, remove "_" before An, 155- change to 375 µL, 163- change to 25 µL; change to 25 µL, 180- change to 100 µL, 181- change to 3000 µL, 183- change to 100 µL, 184- change to 1 mL, 201 the Student's T or, 209- remove "throughout the first year", 214- A p-value of < 0.05, 220- remove "have been detransplanted" put "had grafts removed", 225- prednisone, 227- Cyclosporine, 259- respectively were, 260- remove "respectively", 269- remove "a simplified equation for glomerular filtration rate estimated by modification of diet in renal disease" you defined MDRD-eGFR earlier, 270- measure (typo), 282-308- In these tables you present the numbers as either ",0", ".0" or "0.0". They should all be "0.0" as that is what you are reporting in the text, 326- (Table 6a models), 327- were tested, 329- Table 6 (Table 6b , 332- remove "No variable is transformed.", 339- Table 6a, 354- Table 6b, 354-360- MDRD-pGFR is not in either of these tables. Either insert the equation like you do in Table 8 or remove them from the legend, 365- (Table 4), 366- Table 8, 378- (1Y) post, 379- remove "A simplified equation for glomerular filtration rate estimated by modification of diet in renal disease" you defined MDRD-eGFR prior, 387- significantly, 398- Our results are similar to previous studies, 409- change to "donors, which were considered as controls, and recipients", 417- analyzed, 420- peripheral tissues, indicated , 421- arginine , 423- in arginine metabolism, 424- C-reactive protein, 431- change to "male and female patients who have had complications and those who have not.", 442- results, which , 444- r=0.451, (ρ=0.416 and r=0.451 respectively), 446- ρM3=0.649 , 451- (P=0.017, ρ=0.431 and P=0.010, ρ=-0.450 respectively), 453-456- change to "out. They were models 13, 14 and 15 (Table 6b: P= 0.000, r2 = 0.599, r2adj= 0.549; P= 0.000, r2 455 =0.548, r2adj = 0.497; P= 0.000, r2= 0.553, r2adj = 0.517 respectively)." , 457- Add: "To put these findings in perspective, in many areas", 464- Remove "On another note,". The p-values of these three models are highly significant (P=0.000) and , 466- Also, the p-values for , 468- In addition, model 13, which , 470- One unit , 476- In 2014, a study , 479- measured (typo) GFR at day (typo) 5, 481- indicator of DGF and plasma levels of free thiols at 30 minutes and 90 minutes post-transplantation, 488- blood flow and GFR relatively , 493- Comment: What is the difference between NADPH and molecular NADPH?, 496- an increased asymmetric dimethylarginine (ADMA) , 500- endogenous NO inhibitors, ADMA and , 501- as well as the NO precursors (arginine, citrulline and ornithine) with , 502- were studied. , 506- 25 healthy subjects, NO showed , 516- decreases NO production , 517- of renal allografts, 518- serum nitrate and episodes of acute rejection , 525-526- Remove "The results of the most important of these are reported below." 532-534: Remove the spaces before and after "/" , 535- Remove "estimated glomerular filtration rate", 542- donor's eGFR as , 550- donor eGFR, , 551-554- Remove "This is a simple but useful guide for estimating graft function one year after renal transplantation, particularly in marginal donors (elderly patients, patients with hypertension and atherosclerotic disease), in the clinical setting. , 560- recipient GFR estimated at six , 561- Add: One study attempted to predict the GFR one year , 562- transplantation was carried out in a large , 564- period were (change) analyzed (typo) , 573- They concluded that these variable are considered , 576- trends towards measured GFR , 577- no significant differences were observed, 583- validated in multiple large international , 585- the race-free (typo) 2021 (moved) CKD-EPI equation (developed in individuals with native kidneys) and , 586-587- validation cohorts [62]. The P30 values (P30 being the proportion of eGFR within 30% of measured GFR) ranged from 73.0% to 91.3% [62]. 590- clinical, induction (typo)/primary immunosupressive (typo) treatment, 592- 593- change to: warm ischaemia and MDRD-eGFR at six , References- You removed some citations. Make sure the final number is correct and reflected in the text. On line 587 you have 62 citations. Should be less.

7. PLOS authors have the option to publish the peer review history of their article (what does this mean?). If published, this will include your full peer review and any attached files.

Reviewer #3: No

Reviewer #5: **Yes: **Sean R. Campbell

---

## [Author Response · Author response to Decision Letter 3]

7 Jul 2024

Algiers, July 7th 2024.

Dear Academic Editor and Peer Reviewers,

 The authors would like to thank you for the time taking to reread and improve this manuscript.

We have responded to the three requirements of PLOS ONE and to the requests for corrections from Reviewer n#5.

PLOS ONE requirements 

1- Please note that the content of our financial disclosure was changed in the cover letter (May 9th, 2024). We have removed the reference to personal contributions by Djamila IZEMRANE and Nacim HAMDIS.

2- As far as depositing our laboratory protocols in protocols.io is concerned, we feel that there is no need to make an additional contribution, given that these protocols are referenced (see methods section) and that the analysis methods are common methods with few modifications.

3- We have checked our original list of 56 references in the "Retraction Watch Database" Version 1.0.8.0. No references have been retracted. The final number of references is correct and in accordance with the text.

Response to reviewers

Reviewer#3

The authors thank you for your comment.

Reviewer#5 

The authors thank you for your scrupulous examination of language, words and grammar, as well as for all your recommendations.

1- Typos and grammatical errors in the docuement entitled "Revised Manuscript with Track Changes" have been corrected and hilighted in green or in red : 

line 62- kidney size (we have removed renal), line 85- endothelial cell dysfunction (we have added cell), line 112- we have removed "and ;" and "to", line 113- post-transplantation, line 130- blood samples were, line 142- using the thiobarbituric, we have removed "_" before An, lines 143, 150, 167, 168, 170, and 171 - we have respectively changed 375 µl, 25 µl, 100 µl, 3000 µl, 100 µl and 1 ml to 375 µL, 25 µL, 100 µL, 3000 µL, 100 µL and 1 mL, line 188- the Student's test, line 196- we have removed "throughout the first year", line 201- A p-value of < 0.05, line 207- we have removed "have been detransplanted" and we have put "had grafts removed", line 212- prednisone, line 214- Cyclosporine, line 244 and line 246- respectively were, line 248- we have removed "respectively", line 258- we have removed "a simplified equation for glomerular filtration rate estimated by modification of diet in renal disease", line 259- measure (typo), lines 271-298- In these tables all numbers are presented as "0.0" as indicated in the text, line 315- (Table 6a models), line 316- were tested, line 318- in Table 6 (Table 6b), line 321- we have removed "No variable is transformed.", line 328- Table 6a, line 343- Table 6b, line 344-349- we have removed MDRD-pGFR from the legend, line 354- (Table 4), line 355- Table 8, line 367- (1Y) post, lines 368 and 369- we have removed "A simplified equation for glomerular filtration rate estimated by modification of diet in renal disease", line 375- significantly, line 379- Our results are similar to previous studies, line 384- change to "donors, which were considered as controls, and recipients", line 393- analyzed, line 396- peripheral tissues, indicated , line 397- arginine , line 399- in arginine metabolism, line 400- C-reactive protein, line 407- "male and female patients who have had complications and those who have not.", line 418- results, which , line 420- r=0.451, (ρ=0.416 and r=0.451 respectively), line 422- (rD2=0.528, rD3=0.451, rD6=0.478, ρD8=0.531, rM1=0.563, ρM3=0,649 respectively), line 427- (P=0.017, ρ=0.431 and P=0.010, ρ=-0.450 respectively), line 429-432- change to "out. They were models 13, 14 and 15 (Table 6b: P= 0.000, r2 = 0.599, r2adj= 0.549; P= 0.000, r2 455 =0.548, r2adj = 0.497; P= 0.000, r2= 0.553, r2adj = 0.517 respectively)." , line 433- we have added : "To put these findings in perspective, in many areas", line 440-we have removed "On another note," The p-values, line 442- Also, the p-values for, line 444- In addition, model 13, which, lines 446 and 447- One unit, line 452- In 2014, a study, line 455- measured (typo) GFR at day (typo) 5, lines 457 and 458- indicator of DGF and plasma levels of free thiols at 30 minutes and 90 minutes post-transplantation, line 464- blood flow and GFR relatively, lines 472 and 473- an increased asymmetric dimethylarginine (ADMA), line 476- endogenous NO inhibitors, ADMA and , line 477- as well as the NO precursors (arginine, citrulline and ornithine) with , line 478- were studied, line 482- 25 healthy subjects, NO showed, line 493- decreases NO production, line 494- of renal allografts revealed a significant increase of serum nitrate and episodes of acute rejection , lines 501 and 502- we have removed "The results of the most important of these are reported below." lines 508-510- we have removed the spaces before and after "/", line 511- we have removed "estimated glomerular filtration rate", line 518- donor's eGFR, line 526- donor eGFR, line 527-529- we have removed "This is a simple but useful guide for estimating graft function one year after renal transplantation, particularly in marginal donors (elderly patients, patients with hypertension and atherosclerotic disease), in the clinical setting", line 536- recipient GFR estimated at six months, line 537- Early prediction of GFR one year, line 538- transplantation was carried out in a large , line 540- period were analyzed (typo), line 549- variables, lines 552 and 553- trends towards measured GFR, line 553- no significant differences were observed, line 559- validated in multiple large international, line 561- the race-free 2021 CKD-EPI equation developed in individuals with native kidneys, lines 562 and 563- The P30 values (P30 being the proportion of eGFR within 30% of measured GFR) ranged from 73.0% to 91.3%, line 566- clinical, induction (typo)/primary immunosupressive (typo) treatment, lines 568 and 569- warm ischaemia and MDRD-eGFR at six months. 

 2- On line 469, there is no difference between NADPH and molecular NADPH. We have deleted "and molecular NADPH" from the sentence. We made a mistake in paraphrasing the authors of reference 43.

3- The final number of references is correct (56) and in line with the text. 

Sincerely yours,

Djamila IZEMRANE, Ali BENZIANE and Ahcène BAZ on behalf of the authors.

Corresponding author: Djamila IZEMRANE. 

- Laboratory of Biology and Animal Physiology, Higher Normal School, Kouba, Algiers, Algeria, 16308.

- National Higher Veterinary School, Issad Abbes, Oued Smar, Algiers, Algeria, 16059.

Email: d.izemrane@ensv.dz. izemranedjamila@gmail.com. Phone number: +213 669 20 82 01

---

## [Editor Report · Decision Letter 4]

12 Jul 2024

Living donors kidney transplantation and oxidative stress: nitric oxide as a predictive marker of graft function

PONE-D-23-24980R4

Dear Dr. IZEMRANE,

We’re pleased to inform you that your manuscript has been judged scientifically suitable for publication and will be formally accepted for publication once it meets all outstanding technical requirements.

Kind regards,

John Richard Lee, M.D.

Academic Editor

PLOS ONE
---

## [Editor Report · Acceptance letter]

23 Jul 2024

PONE-D-23-24980R4 

PLOS ONE

Dear Dr. Izemrane, 

I'm pleased to inform you that your manuscript has been deemed suitable for publication in PLOS ONE. Congratulations! Your manuscript is now being handed over to our production team.

Kind regards, 

on behalf of

Dr. John Richard Lee 

Academic Editor

PLOS ONE